# Simple and Efficient Heterogeneous Temporal Graph Neural Network

**Yili Wang**[1,2]**, Tairan Huang**[1]**, Changlong He**[1]**, Qiutong Li**[1]**, Jianliang Gao**[1,*]
[1]Central South University
[2]The Hong Kong University of Science and Technology (Guangzhou)
yiliwang@hkust-gz.edu.cn, gaojianliang@csu.edu.cn

## Abstract

Heterogeneous temporal graphs (HTGs) are ubiquitous data structures in the real world. Recently, to enhance representation learning on HTGs, numerous attention-based neural networks have been proposed. Despite these successes, existing methods rely on a decoupled temporal and spatial learning paradigm, which weakens interactions of spatio-temporal information and leads to a high model complexity. To bridge this gap, we propose a novel learning paradigm for HTGs called **S**imple and **E**fficient **H**eterogeneous **T**emporal **G**raph **N**eural **N**etwork (SE-HTGNN). Specifically, we innovatively integrate temporal modeling into spatial learning via a novel dynamic attention mechanism, which substantially reduces model complexity while enhancing discriminative representation learning on HTGs. Additionally, to comprehensively and adaptively understand HTGs, we leverage large language models to prompt SE-HTGNN, enabling the model to capture the implicit properties of node types as prior knowledge. Extensive experiments demonstrate that SE-HTGNN achieves up to **10× speed-up** over the state-of-the-art and latest baseline while maintaining the best forecasting accuracy.

## 1 Introduction

Heterogeneous temporal graphs (HTGs) have been commonly used to model complex systems in the real world, such as e-commerce networks [1–3], epidemic networks [4, 5], and traffic networks [6, 7]. While static heterogeneous graphs are characterized by diverse node types and relations among connected nodes, HTGs, as shown in Figure 1 (a), extend this data structure by incorporating a temporal dimension. Therefore, learning on HTGs necessitates not only addressing spatial heterogeneity but also capturing the interactions among graph snapshots.

Recently, various heterogeneous dynamic graph neural networks (HDGNNs) have been proposed and have achieved remarkable progress in learning on HTGs [8, 9]. Figure 1 (b) shows a general framework of HDGNNs [10–13], which is characterized by performing spatial and temporal modeling in two sequential stages. Specifically, spatial modeling step employs hierarchical (node- and relation-level) attention-based aggregation on each graph snapshot to generate spatial representations of the target node. Subsequently, a sequence-based module is employed to model the temporal dependencies among these spatial representations, enabling the prediction of future representations for downstream tasks. Such sequence-based module is typically recurrent neural network (RNN) or Transformer [14].

Despite these successes, existing HDGNNs still faced the following limitations: (1) *High model complexity leads to optimization challenges and degrades efficiency.* Specifically, existing methods are incremental improvements upon prior frameworks, rather than breakthroughs, which leads to increasingly complex architectures. For example, stacking additional attention layers and as-

---

*Corresponding Author

39th Conference on Neural Information Processing Systems (NeurIPS 2025).

signing non-shared parameters for each graph snapshot causes the parameter size to grow linearly with the length of the time window, limiting the scalability and efficiency of HDGNNs. (2) *Decoupled spatial and temporal learning weakens the interaction of spatio-temporal information.* From the temporal perspective, the data it receives has already been "compressed" by spatial modeling, rather than comprehensive spatial information, making it difficult to capture global spatio-temporal dependencies of HTGs. From the spatial perspective, attention-based aggregation on each graph snapshot is temporally agnostic, which constrains the receptive field of the attention mechanism and leads to a phenomenon we term attention discontinuity. That is, attention coefficients are computed solely based on the current graph snapshot without referring to historical attention information, making it difficult to capture consistent long-term patterns and increasing the risk of convergence to local optima.

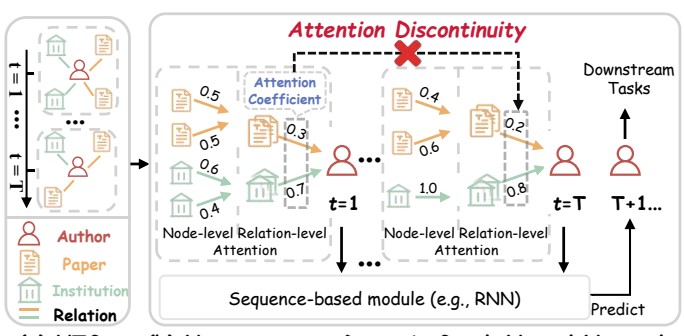

Figure 1: (a) shows a heterogeneous temporal graph (HTG). (b) shows a general framework of attention-based HDGNNs. However, decoupled spatial-temporal modeling strategy results in attention discontinuity (attention on each graph snapshot is computed in isolation), increasing the risk of convergence to local optima.

To bridge this gap, this paper proposes a novel attention-based learning paradigm for HTGs called **S**imple and **E**fficient **H**eterogeneous **T**emporal **G**raph **N**eural **N**etwork (SE-HTGNN). *To address the first challenge,* we redesign the representation learning of HTGs and reduce model complexity from two aspects. At the micro level, we simplify redundant attention layers and linear projections. At the macro level, we integrate temporal modeling into spatial learning to reduce the learning stages. *To address the second challenge,* we propose a novel dynamic attention mechanism to unify spatial and temporal modeling. Specifically, to strengthen the spatio-temporal interactions, the dynamic attention mechanism retains attention coefficients on historical graph snapshots to guide subsequent spatial modeling, thereby improving the overall discriminative representations learning of HTGs. Additionally, to comprehensively and adaptively understand HTGs, we leverage large language models (LLMs [15, 16]) to prompt SE-HTGNN, enabling the model to capture the implicit properties of node types as prior knowledge. We evaluate our method on several real-world HTG datasets across various downstream tasks. Extensive experiments show that SE-HTGNN significantly outperforms the state-of-the-art and latest baselines in both performance and efficiency.

The contributions of this work are summarized below.

- We propose a novel attention-based learning paradigm for HTGs termed SE-HTGNN. By employing a redesigned lightweight architecture and unifying spatial-temporal modeling, SE-HTGNN enables efficient and high-quality representation learning on HTGs.

- We propose a novel dynamic attention mechanism that retains attention information on historical graph snapshots to generate more effective attention coefficients for subsequent graph snapshots. In addition, we introduce LLMs to inject external knowledge into the attention process, thereby enhancing the adaptability and performance of SE-HTGNN.

- Extensive experiments on several real-world datasets demonstrate that SE-HTGNN significantly outperforms the state-of-the-art in performance and efficiency.

## 2   Related Work

**Heterogeneous Graph Neural Networks.** In recent years, there has been explosive growth in heterogeneous graph neural networks (HGNNs) [17–19], driven by the pursuit of improved performance in various applications [20–22]. To address heterogeneity, HGNNs typically employ a hierarchical attention mechanism during aggregation process. MAGNN [23], NIRec [24], and NDS [25] capture neighbor features through a node-level attention (e.g., GAT [26]) and then fuse these features using

a relation-level attention (e.g., HAN [27]). Despite the success of these approaches, hierarchical attention has become a speed bottleneck, limiting further scalability. To enhance efficiency, some studies have sought to simplify HGNN models. SeHGNN [28] was the first to propose the non-necessity of node-level attention, observing that prior methods tend to assign nearly uniform attention to all neighbors. Meanwhile, methods such as MHGCN [29] and RpHGNN [30] demonstrated that a well-designed relation-level attention mechanism can be sufficiently effective. This may be attributed to the fact that intra-type neighbors tend to exhibit lower variance compared to inter-type ones. Therefore, focusing solely on the inter-level (relation-level) attention can unexpectedly yield better performance. Although models designed for static graphs struggle to capture the complex spatio-temporal dependencies in HTGs, these methods inspire us to redesign architectures for HTGs.

**Dynamic Graph Neural Networks (DGNN).** Dynamic graph structures [31, 32] have been extensively explored in the literature, leading to numerous successful applications [33, 7, 34–36]. To generalize the success of DGNN [37–39], there has been considerable research on heterogeneous temporal graph [40, 41]. Specifically, existing heterogeneous DGNNs can be classified into two main categories. **(a)** *compress-based methods:* HGT+ [8] and DHGAS [9], as Transformer [14] variants, compress all graphs into a single graph for efficient representation learning. **(b)** *snapshot-based methods:* DyHATR [11], HTGNN [12], and CasMLN [13] conduct fine-grained spatial learning on each snapshot graph separately, followed by temporal modeling of these spatial representations.

Despite these successes, both types of methods have inherent limitations. For (a) methods, compressing snapshots results in structural information loss and incurs high GPU memory costs, making it difficult to handle large-scale HTGs datasets. For (b) methods, the challenges lie in the slow training speed due to the multiple learning steps (e.g., hierarchical attention and decoupled spatio-temporal modeling) and learnable parameters. Furthermore, both types of methods compute attention coefficients independently at each time step, without referring to historical attention information from previous time steps, which reduces efficiency and increases the risk of convergence to local optima. Beyond this, following the successful application of LLMs in various domains [42–45], the challenge of efficiently leveraging LLMs to advance DGNNs is increasingly relevant.

## 3 Preliminary and Notations

### 3.1 Heterogeneous Temporal Graph (HTG)

Heterogeneous Temporal graph consists of multiple snapshots that evolve over time. Each snapshot is a heterogeneous graph $G = (V, E, X, \mathcal{T}_n, \mathcal{T}_r)$, in which $V$ is the node set and $E$ represents the relation set, $X$ is the feature set. The $\mathcal{T}_n$ and $\mathcal{T}_r$ represent type set of nodes and relations, where $|\mathcal{T}_n| + |\mathcal{T}_r| \geq 2$. Heterogeneous temporal graph $\mathcal{G} = \left( \{G^t\}_{t=1}^T \right)$ is defined as a set of heterogeneous graphs, where $T$ is the number of timestamps, $G^t$ is the snapshot graph at time $t$. Furthermore, $\mathcal{V} = \bigcup_{t=1}^T V^t$, $\mathcal{E} = \bigcup_{t=1}^T E^t$, $\mathcal{X} = \bigcup_{t=1}^T X^t$ are defined as the set of node, relation and feature, respectively.

**HTG Downstream Task.** Without loss of generality, we formulate various downstream tasks on HTGs as a multi-step to multi-step forecasting task. Given a HTGs $\mathcal{G} = \left( \{G^t\}_{t=1}^T, \mathcal{V}, \mathcal{E}, \mathcal{X} \right)$, its downstream task can be formulated as follows:

$$\mathcal{F}(G^{t-(\gamma-1)}, \ldots, G^t; \theta) \rightarrow (\hat{Y}^{t+1}, \cdots, \hat{Y}^{t+\beta}), \tag{1}$$

where $\gamma$ denotes time window size, $\beta$ denotes prediction steps, $\mathcal{F}(\cdot)$ denotes forecasting model, $\theta$ denotes learnable parameter, $\hat{Y}^t$ denotes the predictive value at time step $t$.

### 3.2 Heterogeneous Dynamic GNNs

Recall that the general framework of heterogeneous graph neural networks (HGNNs) always consists of two processes: (I) aggregating intra-relation neighbor representations, (II) fusing inter-relation neighbor representations. For example, given a node $i$, the process of updating its representation $\mathbf{h}_i$ once in HGNN can be formulated as follows:

$$\mathbf{h}_i \leftarrow \underset{\forall r \in \mathcal{T}_r}{\mathbf{HAgg}}(\mathbf{Agg}(\{\mathbf{h}_j : j \in \mathcal{N}_r(i)\})), \tag{2}$$

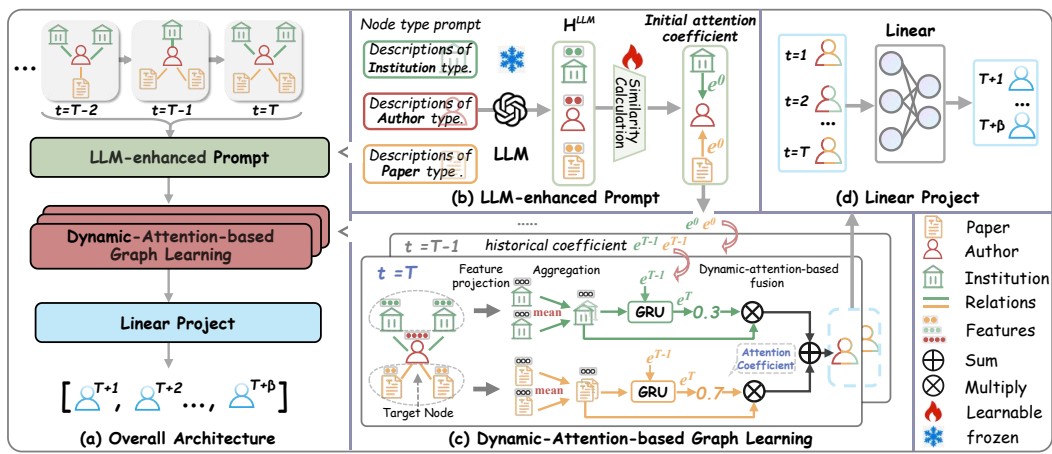

Figure 2: (a) The overall architecture of SE-HTGNN. (b) The LLM-enhanced prompt module takes type prompts as input and generates initial attention coefficients $e^0$ by leveraging LLM-enhanced prior knowledge. (c) The dynamic-attention-based graph learning module predicts current attention coefficients $e^T$ from current features and historical coefficients $e^{T-1}$, enabling effective representations fusion. (d) The linear project module maps representations to future prediction steps for downstream tasks.

where $\mathcal{N}_r(i)$ is the neighbor of node $i$ with the relation type $r$, including itself, $\mathbf{Agg}(\cdot)$ is the message aggregation function that aggregates the intra-relation neighbor node representations, $\mathbf{HAgg}(\cdot)$ is a fuse function which fuses the inter-relation representations. To address heterogeneity, these two functions are designed with node-level and relation-level attention, respectively (e.g., HAN [27]).

Based on the formal definition of HGNN, the previous heterogeneous dynamic graph neural network (HDGNN) was designed by two processes: (I) obtaining spatial representation from each graph snapshot, (II) modeling the temporal dependencies among these representations, enabling the prediction of future representations for downstream tasks. These processes can be formulated as follow:

$$
\begin{aligned}
\mathbf{h}_i^t &\leftarrow \underset{\forall r \in \mathcal{T}_r}{\mathbf{HAgg}}(\mathbf{Agg}(\{\mathbf{h}_j^t : j \in \mathcal{N}_r^t(i)\})), \\
\{\mathbf{h}_i^t\}_{t=T+1}^{T+\beta} &\leftarrow \mathbf{Sequence}(\{\mathbf{h}_i^t : 1 \leq t \leq T\}),
\end{aligned}
\tag{3}
$$

where $\mathcal{N}_r^t(i)$ is the neighbor of node $i$ with the relation type $r$ at timestamp $t$, $\mathbf{h}_i^t$ is the representation of node $i$ at timestamp $t$, $\mathbf{Sequence}(\{\cdot\})$ stands for the sequence-based method (e.g., RNN and Transformer) that can model the temporal dependencies among the representations $\mathbf{h}_i^t$ from different graph snapshots to generate the output $\{\mathbf{h}_i^{T+1}, ..., \mathbf{h}_i^{T+\beta}\}$ for downstream tasks, where $\beta$ denotes prediction steps. We provide a more detailed background in the appendix A.

## 4 Methodology

In this section, we will detail the three clear modules of SE-HTGNN: dynamic-attention-based graph learning, LLM-enhanced prompt, and linear project. The overall architecture is shown in Figure 2.

### 4.1 Dynamic-attention-based Spatial Learning

**Heterogeneous Feature Projection.** As the features of different types of nodes within HTGs have their own feature space, the first priority is to project them into the common feature space using a type-specific projection. This process can be formulated as follow:

$$
\mathbf{H}_v^t = \mathbf{W}_v \cdot \mathbf{X}_v^t + \mathbf{b}_v,
\tag{4}
$$

where $\mathbf{X}_v^t$ denotes raw attribute corresponding node type $v$ at time $t$, $\mathbf{W}_v$ is the trainable type-specific transformation matrix, $\mathbf{b}_v$ is the trainable type-specific bias. Then these heterogeneous features can be aggregated in the same dimension, and the heterogeneity is also preserved.

**Simplified Neighbor Aggregation.** Existing HDGNNs typically employ a node-level attention mechanism for attentive neighbor aggregation. However, considering that intra-type neighbors in HTGs tend to exhibit lower variance compared to inter-type ones, we simplify the neighbor aggregation process to reduce excessive parameters and alleviate optimization difficulties. Specifically, as shown in Figure 2 (c), this paper implements the basic GCN [46] in a non-parametric manner as the aggregation function, which fairly aggregates the neighbor information without introducing redundant linear operations. This process can be formulated as[2]:

$$\mathbf{H}_{v,r}^t = \sigma(\mathbf{A}_r^t \mathbf{H}_{\mathcal{N}_r^t(v)}^t), \tag{5}$$

where $\mathbf{H}_{v,r}^t$ denotes intermediate representations updated by neighbor under relation $r$ at time $t$, $\mathcal{N}_r^t(\cdot)$ denotes node type of neighbor under relation $r$ at time $t$, $\mathbf{A}_r^t$ is the normalized adjacency matrix corresponding to relation $r$ at time $t$, $\sigma$ is the activation function (i.e., ELU function). After aggregating different types of neighbors separately, we collected various types of intermediate representations $\{\mathbf{H}_{v,r}^t | r \in \mathcal{R}(v)\}$, where $\mathcal{R}(v)$ denotes the set of relations with $v$ as their target node type. Next, we need to fuse these intermediate representations to generate the final representations.

**Dynamic-Attention-based Fusion.** Since the decoupled spatio-temporal modeling strategy limits the receptive field of attention mechanisms, previous methods often converge to suboptimal performance. In view of this limitation, we propose the dynamic attention mechanism to fuse representations.

Specifically, to incorporate temporal information into spatial learning, dynamic attention utilizes the gate recurrent unit (GRU) to produce coefficients in a sequential manner. In this way, historical attention coefficients are stored in the hidden state of GRU, which can guide the attention calculation on the subsequent snapshots. Additionally, since the evolving trends may differ across different relations in HTGs, we use relation-wise GRUs to independently capture these various trends. As shown in Figure 2 (c), at each time $t$, this module takes the current intermediate representation $\mathbf{H}_{v,r}^t$ and the historical coefficients $\mathbf{e}_{v,r}^{t-1}$ as input to predict the current attention coefficients $\mathbf{e}_{v,r}^t$. Finally, these attention coefficients are averaged and normalized. This process can be formulated as:

$$\underbrace{\overbrace{\mathbf{e}_{v,r}^t}^{\text{attention coefficient}}}_{\text{hidden state}} = \text{GRU}_r(\ \underbrace{\overbrace{\mathbf{H}_{v,r}^t}^{\text{representation}}}_{\text{input}}\ ,\ \underbrace{\overbrace{\mathbf{e}_{v,r}^{t-1}}^{\text{historical coefficient}}}_{\text{hidden state}}\ ), \tag{6}$$

$$\alpha_r^t = \frac{\exp\left(\overline{\mathbf{e}}_{v,r}^t\right)}{\sum_{r' \in \mathcal{R}(v)} \exp\left(\overline{\mathbf{e}}_{v,r'}^t\right)}, \tag{7}$$

where $\mathbf{e}_{v,r}^t \in \mathbb{R}^{n \times 1}$ denotes hidden state that contain attention coefficient of relation $r$ to node type $v$ at time $t$, $\overline{\mathbf{e}}_{v,r}^t \in \mathbb{R}^1$ denotes the average of $\mathbf{e}_{v,r}^t$, $\alpha_r^t \in \mathbb{R}^1$ denotes normalized coefficient, $\mathcal{R}(v)$ denotes the set of relations with $v$ as their target node type, $\text{GRU}_r(\cdot)$ denotes relation-wise GRU corresponding to relation $r$. Notably, the initial hidden state $\mathbf{e}_{v,r}^0$ of GRU is essential as it can significantly impact the model's convergence speed and overall performance. Therefore, we employ Large Language Models (LLMs) to provide external knowledge, thereby initializing $\mathbf{e}_{v,r}^0$ with more meaningful vectors. The detailed initialization of $\mathbf{e}_{v,r}^0$ will be introduced in Section 4.2. Additionally, dynamic attention is feasible to extend it into multi-head attention by extending the dimension of the hidden state $\mathbf{e}_{v,r}^t$ from $\mathbb{R}^{n \times 1}$ to $\mathbb{R}^{n \times k}$, where $k$ is the number of heads. For simplicity, we only discuss the one-head here. After obtaining the attention coefficient of each relation, we fuse these intermediate representations to generate the final representations for the target node. This process can be formulated as:

$$\mathbf{H}_v^t = \sum_{r \in \mathcal{R}(v)} \alpha_r^t \cdot \mathbf{H}_{v,r}^t \tag{8}$$

By performing such graph learning on each graph snapshot, we obtain spatio-temporal representations at each time step $\{\mathbf{H}_v^t | 1 \le t \le T\}$, where $T$ denotes the number of snapshots. Next, we need to project these representations into the desired future steps for downstream tasks.

## 4.2 LLM-enhanced Prompt

In the above sections, this paper integrates temporal information into the attention mechanism using Gated Recurrent Units (GRUs). Since a well-initialized hidden state for GRUs enhances the model's

---

[2]To simplify notations, we omit the layer superscript.

ability to more effectively understand the context of data and improve overall performance [47], this paper proposes a novel initialization method based on large language models (LLMs). As shown in Figure 2 (b), for a given node type $v$, we first construct a node type prompt consisting of two elements: (I) a concise text-based description of type $v$, and (II) an instruction requiring the LLMs to output in a fixed format. Then the node type prompt is processed by the LLMs (GPT-3.5 [48] or Llama3 [16]), which facilitates the extraction of implicit information of HTGs from sequential text. We use the embeddings of the final hidden layer of the LLMs as the semantic representations of node types, which encode prior knowledge and domain understanding. This process can be formulated as:

$$\text{Prompt}(v) = \{\text{Introduction to type } v; \text{ Instruction.}\}, \tag{9}$$

$$\mathbf{H}_v^{LLM} = \text{LLM}(\text{Prompt}(v)), \ \forall v \in \mathcal{T}_n \tag{10}$$

where $\mathbf{H}_v^{LLM}$ denotes the LLM-based representation of the node type $v$ augmented through the knowledge by the LLM. Considering that the hidden state $e_{v,r}^0$ needs to be initialized as the initial attention coefficient , we initialize $e_{v,r}^0$ based on the similarity calculation of $\mathbf{H}_u^{LLM}$ and $\mathbf{H}_v^{LLM}$ corresponding to source and target node types $u$ and $v$ of relation $r$. This process can be formulated as follows:

$$\mathbf{Q}_u = \mathbf{W}_Q \mathbf{H}_u^{LLM}, \mathbf{K}_v = \mathbf{W}_K \mathbf{H}_v^{LLM}, \beta_r = \mathbf{Q}_u \mathbf{K}_v^\top \tag{11}$$

$$\mathbf{e}_{v,r}^0 = \frac{\exp(\beta_r)}{\sum_{r' \in \mathcal{R}(v)} \exp(\beta_{r'})}, \tag{12}$$

where $e_{v,r}^0$ is the initial attention coefficient as described in Section 4.1, and $\mathbf{W}_Q, \mathbf{W}_K$ are the learnable transformation matrices. It is worth noting that since our attention is applied at the relation level, the number of prompts processed by the LLM depends on the number of node types in the HTGs, rather than the total number of nodes, leading to efficient computation. Furthermore, the LLM processing step can be performed during preprocessing to reduce memory overhead during training. In implementation, we employ LLaMA3-8B [16] to enhance our model. We also provide concrete prompt examples and compare the performance of different LLMs in the appendix B.

## 4.3 Linear Project

Since our model integrates temporal information into the graph learning step, the final step only involves projecting these obtained spatial-temporal representations into the desired number of future steps, without the requirement for additional temporal modeling. This makes the overall process more streamlined and computationally efficient. As shown in the figure 2 (d), given spatial-temporal representations over T time steps $\mathbf{Z}_v = [\mathbf{H}_v^1, ..., \mathbf{H}_v^T] \in \mathbb{R}^{N \times d \times T}$, a linear transformation is utilized to project them into $\beta$ future steps. Here, $N$ denotes the number of nodes corresponding to type $v$, and $d$ represents the dimension of the representations. This process can be formulated as follows:

$$\mathbf{Z}_v' = \mathbf{Z}_v \cdot \mathbf{W} + \mathbf{b}, \tag{13}$$

where $\mathbf{Z}_v' = [\mathbf{H}_v^{T+1}, ..., \mathbf{H}_v^{T+\beta}] \in \mathbb{R}^{N \times d \times \beta}$ denotes the predicted representations for downstream tasks, $\mathbf{W} \in \mathbb{R}^{T \times \beta}$ and $\mathbf{b}$ is the trainable matrix of linear transformation.

## 4.4 Optimization

We then pass the yielded representation of node type $v$ to a two-layer multilayer perceptron (MLP) to capture non-linear interactions between the representations. Take the $T + 1$ as an example:

$$\mathbf{H}_v = \text{MLP}(\mathbf{H}_v^{T+1}) \tag{14}$$

Next, we introduce loss functions over HTGs. **For link prediction task**, we use the following loss:

$$\mathcal{L} = -\sum_{(i,j) \in \Omega^+} \log \sigma\left(\mathbf{h}_i^\top \mathbf{h}_j\right) - \sum_{(u',v') \in \Omega^-} \log \sigma\left(-\mathbf{h}_{i'}^\top \mathbf{h}_{j'}\right), \tag{15}$$

where $\Omega^+$ and $\Omega^-$ denote the set of observed positive and negative pairs, respectively, and $\sigma$ denotes the sigmoid function. $\mathbf{h}_i, \mathbf{h}_j, \mathbf{h}_{i'}, \mathbf{h}_{j'} \in \mathbf{H}_v$ are node representation vectors output by the MLP. **For**

Table 1: The overall results for different methods for various tasks, including the results of link prediction, classification and node regression. The best and second-best results are shown in **red** and **blue**, respectively. OOM: out of GPU memory.

| Type | Method | Link Prediction | | | | Node Classification | | Node Regression | |
|---|---|---|---|---|---|---|---|---|---|
| | | OGBN-MAG | | Aminer | | YELP | | COVID-19 (30-day) | |
| | | (AUC%)↑ | (AP%)↑ | (AUC%)↑ | (AP%)↑ | (Macro-F1%)↑ | (Recall%)↑ | (MAE)↓ | (RMSE)↓ |
| Homogeneous GNNs | GCN *(ICLR'2017)* | 78.16 ± 1.32 | 76.48 ± 1.86 | 73.12 ± 0.46 | 72.96 ± 0.48 | 37.21 ± 0.52 | 38.02 ± 0.69 | 841 ± 98 | 1497 ± 132 |
| | GAT *(ICLR'2018)* | 79.97 ± 1.98 | 77.62 ± 1.68 | 81.56 ± 1.25 | 79.56 ± 1.36 | 35.39 ± 1.20 | 35.61 ± 1.45 | 814 ± 95 | 1531 ± 219 |
| | TGAT *(ICLR'2020)* | OOM | OOM | 85.69 ± 0.75 | 85.04 ± 0.68 | 39.23 ± 0.78 | 39.87 ± 0.83 | OOM | OOM |
| Heterogeneous GNNs | RGCN *(ESWC'2018)* | 81.25 ± 1.99 | 80.24 ± 1.79 | 82.12 ± 0.12 | 81.24 ± 0.33 | 37.86 ± 0.89 | 38.21 ± 0.48 | 830 ± 85 | 1602 ± 121 |
| | RGAT *(ICLR'2019)* | 87.54 ± 1.12 | 87.09 ± 1.05 | 85.32 ± 0.74 | 84.82 ± 0.86 | 37.74 ± 0.94 | 37.78 ± 1.44 | 785 ± 76 | 1501 ± 94 |
| | HGT *(WWW'2020)* | 84.38 ± 1.22 | 83.99 ± 1.46 | 78.81 ± 1.29 | 77.98 ± 1.62 | 34.28 ± 1.08 | 35.32 ± 0.89 | 799 ± 82 | 1554 ± 98 |
| | DiffMG *(KDD'2021)* | 87.98 ± 1.26 | 87.77 ± 1.20 | 85.14 ± 0.32 | 84.56 ± 0.28 | 39.32 ± 0.89 | 38.85 ± 1.02 | 621 ± 58 | 1286 ± 69 |
| Heterogeneous Dynamic GNNs | DyHATR *(ECML'2021)* | 89.49 ± 0.65 | 86.24 ± 0.91 | 86.46 ± 1.28 | 86.22 ± 1.14 | 38.89 ± 0.64 | 40.98 ± 0.98 | 668 ± 56 | 1322 ± 74 |
| | HGT+ *(WWW'2020)* | OOM | OOM | 85.88 ± 0.38 | 84.24 ± 0.48 | 38.33 ± 0.60 | 40.29 ± 0.56 | OOM | OOM |
| | HTGNN *(SDM'2022)* | **91.21 ± 0.77** | 89.18 ± 1.24 | 85.92 ± 0.93 | 83.74 ± 0.85 | 36.65 ± 1.13 | 38.76 ± 1.22 | 555 ± 34 | 1136 ± 65 |
| | DHGAS *(AAAI'2023)* | OOM | OOM | 88.13 ± 0.90 | 86.92 ± 0.78 | 41.99 ± 1.80 | 42.29 ± 1.25 | **536 ± 43** | **1112 ± 43** |
| | CasMLN *(SIGIR'2024)* | 90.85 ± 0.64 | **89.47 ± 0.57** | **88.53 ± 0.27** | **87.25 ± 0.63** | **42.21 ± 0.51** | **42.57 ± 0.69** | 544 ± 18 | 1119 ± 12 |
| | **SE-HTGNN *(Ours)*** | **93.13 ± 0.56** | **92.71 ± 0.52** | **91.08 ± 0.59** | **90.03 ± 0.48** | **44.24 ± 0.88** | **44.68 ± 0.43** | **497 ± 5** | **1069 ± 11** |
| | Improve. | +2.11% | +3.62% | +2.89% | +3.19% | +4.81% | +4.96% | +7.27% | +4.56% |

**node classification**, the node representations will be projected by MLP to the hidden dimension corresponding to the number of classes. Then the MLP is trained to minimize the cross-entropy loss:

$$\mathcal{L} = -\sum_{i \in \mathcal{V}} \sum_{c=1}^{|\mathcal{T}_n|} y_i[c] log(\hat{y}_i[c]), \tag{16}$$

where $\mathcal{V}$ denotes the set of labeled nodes, $\mathcal{T}_n$ denotes the node type set, $y_i$ is a one-hot vector indicating the label of node $i$, $\hat{y}_i = \text{softmax}(\mathbf{h}_i)$ is the predicted label for the corresponding node. **For node regression**, we use the mean absolute error (MAE) loss as follows:

$$\mathcal{L} = \frac{1}{|\mathcal{V}_L|} \sum_{i \in \mathcal{V}_L} |y_i - \hat{y}_i|, \tag{17}$$

where $\mathcal{V}_L$ is the set of target node, $y_i$ is the ground-truth label (an integer) of node $i$, and $\hat{y}_i = \mathbf{h}_i$ is the regression value node $i$ output of the MLP.

## 5 Experiment

In this section, we present a comprehensive set of experiments to demonstrate the effectiveness of SE-HTGNN. The details about experimental setups are recorded in the appendix C.

### 5.1 Experimental Setups

**Dataset.** We follow the dataset and splits provided by previous works [12, 9]. Specifically, we utilized two link prediction datasets: OGBN-MAG and Aminer, one node classification dataset YELP, and one node regression dataset COVID-19. We repeat all experiments 5 times and report the average results and standard deviations. The description of the datasets are summarized in the appendix C.1.

**Baseline.** We compare the proposed SE-HTGNN with 12 strong baselines of three categories as follow: **(1) Homogeneous GNNs:** GCN [46], GAT [26] and TGAT [37]. These models ignore the heterogeneity of HTGs. **(2) Heterogeneous GNNs:** RGCN [17], RGAT [49], HGT [8], and DiffMG [19]. These models ignore the evolution of HTGs. **(3) Heterogeneous Dynamic GNNs:** DyHATR [11], HGT+ [8], HTGNN [12], DHGAS [9] and CasMLN [13].

### 5.2 Experimental Results

**Link Prediction & Node Classification Task**. For the link prediction task, we use the area under the ROC curve (AUC) and average Precision (AP) as evaluation metrics. For the node classification task, we use Macro-F1 and Recall as evaluation metrics. Macro-F1 score represents the unweighted mean of the F1-score for each label. While the Recall score reflects a model's ability to cover true positives. From the results shown in Table 1, we have the following observations: (1) HDGNNs generally

outperform GNNs, demonstrating the importance of incorporating temporal information when dealing with HTGs. Among them, the attention-based methods CasMLN, performed well on multiple datasets, demonstrating the potential of the attention mechanism. (2) Both HGT+ and DHGAS, which are variants of the Transformer model, are unable to handle large-scale temporal graphs, OGBN-MAG, and long-term graph COVID-19 due to their significant GPU memory consumption. (3) **SE-HTGNN** achieves the best performance by a substantial margin, with an average improvement of approximately 3% in evaluation metrics over the state-of-the-art baseline. This result demonstrates that by using a dynamic attention mechanism, SE-HTGNN can better utilize temporal information to generate superior representations for complex link prediction and node classification tasks.

**Node Regression Task (Long-term).** For the node regression task, we use the mean absolute error (MAE) and root mean square error (RMSE) as evaluation metrics. Our objective on the COVID-19 dataset is to forecast the new daily cases. We report the results in Table 1,2 and observe the following findings: (1) SE-HTGNN again achieves the best performance across all baselines, which demonstrate that our method can adaptively handle various applications of HTGs.

Table 2: Long-term prediction on COVID-19.

| COVID-19 | 60-day prediction | | 90-day prediction | |
|---|---|---|---|---|
| Metric | (MAE)↓ | (RMSE) ↓ | (MAE)↓ | (RMSE)↓ |
| HTGNN | **901 ± 35** | **1787 ± 62** | 1105 ± 26 | 2250 ± 42 |
| DHGAS | 1351 ± 82 | 2809 ± 142 | 1692 ± 108 | 3708 ± 241 |
| CasMLN | 914 ± 52 | 1792 ± 102 | **1084 ± 36** | **2211 ± 48** |
| **SE-HTGNN** | **825 ± 8** | **1701 ± 15** | **1001 ± 12** | **2131 ± 31** |
| Improve. | +8.44% | +4.81% | +6.97% | +3.62% |

(2) As shown in Table 2, under long-term prediction settings (i.e., 60-day and 90-day horizons) on the COVID-19 dataset, all methods experience varying degrees of performance degradation due to the reduced training set and extended prediction length. Nevertheless, our method consistently outperforms the best-performing baseline across all evaluations, achieving an improvement of approximately 7% to 8% in MAE. This superior performance can be attributed to our unified spatial-temporal learning paradigm, which enables the model to better capture long-term temporal dependencies in HTGs compared to decoupled approaches.

Table 3: Ablation study results on various variants of SE-HTGNN.

| Dataset | OGBN-MAG | | Aminer | | YELP | | COVID-19 | |
|---|---|---|---|---|---|---|---|---|
| Metric | (AUC%) ↑ | (AP%) ↑ | (AUC%)↓ | (AP%)↓ | (Macro-F1%)↓ | (Recall%)↓ | (MAE)↓ | (RMSE)↓ |
| w/o LLM$_{Random}$ | 90.87 ± 1.24 | 90.06 ± 1.27 | 87.91 ± 1.54 | 87.05 ± 189 | 41.05 ± 0.93 | 41.29 ± 0.68 | 542 ± 28 | 1181 ± 46 |
| w/o LLM$_{Average}$ | 91.18 ± 0.81 | 89.83 ± 1.29 | 89.76 ± 0.34 | 88.56 ± 0.33 | 43.39 ± 0.62 | 43.82 ± 0.78 | 521 ± 22 | 1102 ± 35 |
| w/o LLM$_{Zero}$ | 91.78 ± 0.68 | 91.08 ± 0.65 | 89.98 ± 0.62 | 88.93 ± 0.76 | 43.31 ± 0.79 | 43.76 ± 1.07 | 524 ± 9 | 1114 ± 19 |
| w/o Att$_{proj}$ | 86.83 ± 1.21 | 86.29 ± 1.35 | 85.42 ± 0.98 | 84.86 ± 1.12 | 38.19 ± 1.56 | 37.82 ± 1.42 | 574 ± 52 | 1222 ± 59 |
| w/o Att$_{self}$ | 91.65 ± 1.22 | 90.65 ± 1.14 | 88.73 ± 0.82 | 88.21 ± 0.94 | 42.41 ± 1.32 | 42.73 ± 0.97 | 545 ± 33 | 1114 ± 42 |
| w/o Att$_{gate}$ | 87.94 ± 2.42 | 87.24 ± 1.89 | 87.42 ± 1.24 | 86.55 ± 1.33 | 38.96 ± 3.27 | 39.26 ± 2.27 | 574 ± 45 | 1216 ± 68 |
| w/o Agg$_{none}$ | 83.91 ± 1.20 | 82.60 ± 1.16 | 62.47 ± 1.23 | 64.91 ± 1.18 | 35.27 ± 2.79 | 35.56 ± 2.52 | 672 ± 78 | 1336 ± 131 |
| w/o Agg$_{gcn}$ | 90.57 ± 0.86 | 92.28 ± 0.75 | 88.40 ± 2.64 | 87.26 ± 2.21 | 43.69 ± 0.79 | 43.82 ± 0.62 | 508 ± 10 | 1080 ± 18 |
| w/o Agg$_{gat}$ | 91.93 ± 0.46 | 89.54 ± 0.48 | 88.37 ± 0.28 | 86.89 ± 0.61 | 42.11 ± 1.32 | 42.23 ± 1.41 | 605 ± 88 | 1231 ± 137 |
| **SE-HTGNN** | **93.13 ± 0.56** | **92.71 ± 0.52** | **91.08 ± 0.59** | **90.03 ± 0.48** | **44.24 ± 0.88** | **44.68 ± 0.43** | **497 ± 5** | **1069 ± 11** |

**Ablation Studies.** In this section, we compare SE-HTGNN with its variants to validate the effectiveness of each component. The description of the variants is given as follow: **w/o LLM** indicates the removal of the LLM-enhanced prompt module, **w/o Att** indicates the removal of the dynamic attention mechanism, and **w/o Agg** indicates the removal of the simplified neighbor aggregation. Furthermore, we use subscripts to denote alternative methods adopted in the ablation experiments: For the w/o LLM, we replace the original LLM initialization with *Random, Average, Zero* initializations. For the w/o Att, we substitute it with *Projected-attention, Self-attention, Gated-attention*. For the w/o Agg, we replace the simplified neighbor aggregation with *None-aggregation, GCN, GAT*.

From the results shown in Table 3, we have the following observation: (1) w/o LLM exhibited a decline in performance, demonstrating the effectiveness of LLMs in HTG representation learning. Among the alternatives, random initialization leads to the most significant degradation, which can be attributed to the noise introduced by the semantically void initialization strategy. (2) w/o Att led to a dramatic collapse in performance, demonstrating the indispensable role of the dynamic attention mechanism (particularly in the context of dynamic graph mining). (3) w/o Agg also shows a clear decrease in performance. This is because traditional aggregation approaches such as GCN and GAT learn parameterized transformations that fail to adapt to the shifting feature distributions in HTGs, thereby adversely affecting the overall model performance. (4) SE-HTGNN consistently outperforms all its variants in the three datasets, validating the effectiveness of each component.

## 5.3 Additional Analysis and Discussion

**Study on Dynamic Attention.** We further investigate the impact of different sequence-based modules on the dynamic attention mechanism. Specifically, we replace the GRU with other modules as follows: LSTM [50], Transformer [14], and Mamba [51]. According to the experimental results shown in Table 4, the original model and the LSTM-based variant achieve comparable performance. However, GRU offers better computational efficiency than LSTM. For Mamba model, as reported in its original paper, it suffers from unstable training. For Transformer model, it is worth noting that it cannot benefit from the LLM-enhanced prompt module, which may account for its inferior performance.

Table 4: Study result on various variants of dynamic attention mechanism.

| Dataset | OGBN-MAG | | Aminer | | YELP | | COVID-19 | |
|---------|----------|----------|----------|----------|----------|----------|----------|----------|
| Metric | (AUC%) ↑ | (AP%) ↑ | (AUC%)↓ | (AP%)↓ | (Macro-F1%)↓ | (Recall%)↓ | (MAE)↓ | (RMSE)↓ |
| LSTM | 92.77 ± 0.48 | 92.28 ± 0.45 | 90.52 ± 0.48 | 89.87 ± 0.58 | 44.31 ± 0.97 | 44.73 ± 0.52 | 506 ± 7 | 1078 ± 16 |
| Mamba | 91.79 ± 1.32 | 90.92 ± 1.45 | 87.81 ± 1.65 | 87.12 ± 1.59 | 41.45 ± 2.41 | 41.58 ± 3.10 | 601 ± 67 | 1259 ± 84 |
| Transformer | 90.75 ± 0.32 | 89.93 ± 0.34 | 90.93 ± 0.45 | 89.89 ± 0.32 | 43.59 ± 1.21 | 43.64 ± 1.19 | 529 ± 12 | 1098 ± 23 |
| **SE-HTGNN** | **93.13 ± 0.56** | **92.71 ± 0.52** | **91.08 ± 0.59** | **90.03 ± 0.48** | **44.24 ± 0.88** | **44.68 ± 0.43** | **497 ± 5** | **1069 ± 11** |

**Efficiency Analysis.** Firstly, we theoretically analyze the time complexity of SE-HTGNN compared to HTGNN and CasMLN, as Table 5 shows. Since DHGAS is an automated architecture search framework, we only report its actual training time. For a concise and fair comparison, all models are simplified to a single-layer structure without common components such as heterogeneous feature projection. We assume there are $T$ time slices for single-step forecasting and $R$ node types, with each node type containing an average of $n$ nodes. The hidden dimension is denoted as $d$, and each relation involves an average of $e$ neighboring nodes per node. It can be observed that existing methods incur $O(d^2)$ complexity in spatial learning due to complex node-level attention and redundant linear projections. Additionally, HTGNN incurs $O(T^2)$ complexity in temporal modeling because of its self-attention mechanism. In contrast, SE-HTGNN achieves significantly lower complexity through a lightweight design and dynamic attention mechanisms that unify spatial and temporal modeling.

To validate our theoretical analysis, we conduct experiments to compare the GPU time consumption. We compare SE-HTGNN with these representative models to evaluate model efficiency. To ensure fairness, the time consumed during the LLM preprocessing stage is also included in the total computation time. As shown in Table 6, SE-HTGNN significantly outperforms all previous methods in terms of actual training time. Specifically, SE-HTGNN achieves a 2.7× speedup over the SOTA on the large-scale dataset OGBN-MAG, and nearly a 10× speedup on the long-sequence dataset COVID-19, which reflects the superiority of SE-HTGNN on the training speed.

Table 5: Time complexity comparison.

| Method | Spatial Learning | Temporal Learning |
|--------|------------------|-------------------|
| HTGNN | $O(TR(ned + nd^2))$ | $O(n(T^2d + Td^2))$ |
| CasMLN | $O(TR(ned + nd^2))$ | $O(Tnd)$ |
| **SE-HTGNN** | $O(\underbrace{TRned}_{\text{Aggregation}} + \underbrace{TRnd}_{\text{Attention}} + \underbrace{Tnd}_{\text{Project}} + \underbrace{Rd'd}_{\text{LLM}}) = O(\underbrace{TRned}_{\text{Total}})$ | |

Table 6: Training cost compared in GPU second.

| Dataset | OGBN-MAG | Aminer | YELP | COVID-19 |
|---------|----------|--------|------|----------|
| HTGNN | 1132 | 472 | 165 | 1403 |
| DHGAS | OOM | 351 | 164 | 11,120 |
| CasMLN | 272 | 135 | 88 | 637 |
| **SE-HTGNN** | **102** | **84** | **47** | **64** |

**Convergence Analysis.** In this section, we delve into the convergence speed analysis of SE-HTGNN with other methods on Aminer and OGBN-MAG datasets. As depicted in Figure 3, our proposed SE-HTGNN achieves convergence within 50 epochs. Meanwhile, HTGNN and CasMLN require more epochs to achieve the best performance. This demonstrates that the unified spatio-temporal modeling strategy of SE-HTGNN not only improves performance but also accelerates convergence.

**Hyper-parameter Sensitive Study.** In this section, we investigate the impact of the time window and embedding dimension on the performance of the model. (1) *Time window size* determines the number of snapshots that can be reviewed during prediction. We validate the effect of size by ranging it from 2 to 9 for three datasets. The results are shown in the left part of Figure 4. We can see that a large size on Aminer and YELP boosts the performance as more historical information is included. However, the time window size has a minor impact on the OGBN-MAG, possibly because each snapshot contains sufficient information. (2) *embedding dimension* refers to the size of the model's hidden representations. The right part of Figure 4 exhibits a rising trend in performance as the embedding

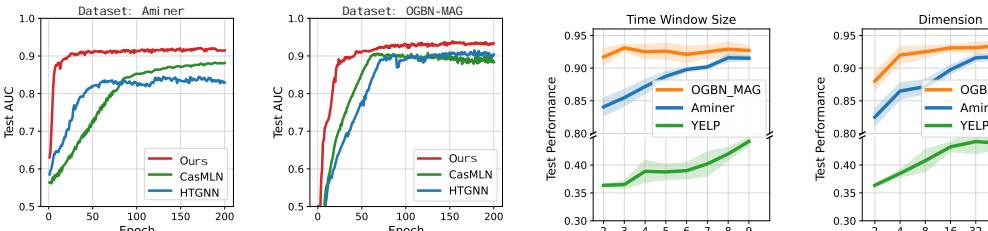

Figure 3: Comparison of convergence speed.    Figure 4: Hyper-parameter sensitive study.

dimension increases, followed by a stabilization phase. To strike a balance between accuracy and efficiency, we set the embedding dimension to 32 for all datasets.

**Visualization and Discussion.** To gain more intuitive insights into our model, we visualize the model's predictions and the dynamic attention coefficients on the COVID-19 dataset. (1) *For the visualization of prediction* (left part of Figure 5), we average the predicted values across all nodes and compare them with the ground truth. It can be observed that existing methods, limited by their decoupled modeling strategy, can only capture coarse-grained trends and fail to reflect fine-grained temporal variations. In contrast, our model produces predictions that closely match

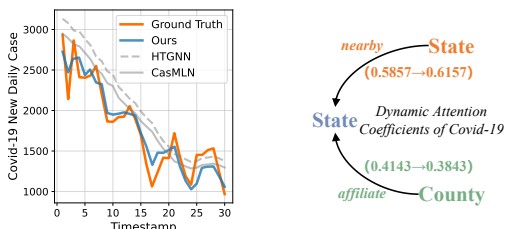

Figure 5: Visualization on Covid-19 dataset.

the ground truth, demonstrating superior capability in capturing subtle temporal dynamics in HTGs.

(2) *For the visualization of dynamic attention coefficients*, the right part of Figure 5 illustrates how attention coefficients evolving over a 30-day period for a specific U.S. state in the COVID-19 dataset. In this figure, the blue "state" denotes a specific state node, the orange "state" refers to all neighboring state-type nodes of the blue state, and the "county" refers to all county-type nodes governed by the blue state. In the task of predicting the new cases in that state, we first observe that the attention coefficients assigned to neighboring states are consistently higher than those for county. This suggests that the state's case increases are likely more influenced by adjacent states. Furthermore, the coefficients of the state-type node increases from 0.58 to 0.61, indicating that recent information from the neighboring state becomes more important than historical data. This observation also supports our insight that historical attention patterns are meaningful for future learning.

**Limitations and Future Work.** This work introduces large language models (LLMs) to enhance the overall performance of model. However, LLM-generated embeddings typically have high and fixed dimensionality. As a result, the learnable transformation matrices required for linear dimensionality reduction contribute a considerable parameter overhead, which can compromise computational efficiency. In future work, we plan to explore dimensionality reduction techniques (e.g., low-rank decomposition and principal component analysis) to compress LLM outputs, aiming to further improve model efficiency and scalability without substantially sacrificing semantic information.

## 6 Conclusion

This paper revisits the existing learning paradigm for HTGs and identifies its limitations. Specifically, existing methods rely on a decoupled temporal and spatial modeling paradigm, which weakens interactions of spatio-temporal information and leads to a high model complexity. To address these limitations, we propose a novel attention-based learning paradigm for HTGs called SE-HTGNN. To the best of our knowledge, SE-HTGNN is the first to propose the concept of a dynamic attention mechanism to unify spatial and temporal modeling, with its innovation lying in utilizing historical attention information to guide subsequent attention processes, thereby improving the model efficiency and performance. Furthermore, we innovatively introduce LLMs to provide external knowledge to improve the adaptability and performance of our model. Extensive experiments on several real-world datasets demonstrate the superiority of SE-HTGNN in both efficiency and performance.

## Acknowledgments and Disclosure of Funding

The work is supported by the National Natural Science Foundation of China (No. 62272487).

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

**Appendix Contents:**

# A  Background

As illustrated in Section 3.2, we provide a general framework for heterogeneous graph neural networks (HGNNs) and hierarchical attention mechanisms. Here, we provide a detailed implementation of this mechanism. In general, the hierarchical attention mechanism consists of node-level attention (e.g., GAT [26]) and relation-level attention (e.g., HAN [27]).

## A.1  Node-level attention

Previous methods typically use GAT [26] to calculate the importance of each neighbor to the central node, thereby aggregating neighbors discriminatively. The attention coefficients are computed using a shared attention parameter $\mathbf{a} \in \mathbb{R}^{2d'}$. For nodes $u$ and $v$, the attention coefficient $e_{uv}$ is given by:

$$e_{uv} = \text{LeakyReLU}\left(\mathbf{a}^T[\mathbf{W}\mathbf{h}_u \,\|\, \mathbf{W}\mathbf{h}_v]\right), \tag{18}$$

$$\alpha_{uv} = \frac{\exp(e_{uv})}{\sum_{v' \in \mathcal{N}(u)} \exp(e_{uv'})} \tag{19}$$

where $\mathbf{h}_u, \mathbf{h}_v$ is the feature vector of node $u$ and $v$, respectively, $\mathbf{W} \in \mathbb{R}^{d \times d'}$ denotes weight matrix, where $d'$ is the dimension of the transformed features. $\|$ denotes concatenation and $\text{LeakyReLU}(x) = \max(0.01x, x)$ denotes activation function. To handle heterogeneity, these methods assign different attention parameterizations for neighbor under different relation types.

## A.2  Relation-level attention

Relation-level attention, also known as semantic-level attention, functions to fuse these representations after collecting information from different types of neighbors [49, 27, 28, 30]. This type of attention does not involve pairwise calculations, making it much more efficient than node-level attention. For each relation $r$, the attention coefficients $\beta_r$ are computed as follows:

$$\beta_r = \frac{\exp\left(\text{MLP}(\mathbf{h}_{u,r})\right)}{\sum_{r' \in \mathcal{R}} \exp\left(\text{MLP}(\mathbf{h}_{u,r'})\right)}, \tag{20}$$

where $\mathbf{h}_{u,r}$ denotes the feature vector updated by neighbor under relation $r$, $\text{MLP}(\cdot)$ denotes multi layer perception with output dimension $d \in \mathbb{R}^1$.

## A.3  Existing Attention Mechanism in Heterogeneous Dynamic GNNs (HDGNNs)

When extending the hierarchical attention mechanism to HDGNNs [10, 11, 13, 12], previous work often employed attention mechanisms in each graph snapshot independently. As an example of relation-level attention, Eq.(20) is extended as follows:

$$\beta_r^t = \frac{\exp\left(\text{MLP}^t(\mathbf{h}_{u,r}^t)\right)}{\sum_{r' \in \mathcal{R}} \exp\left(\text{MLP}^t(\mathbf{h}_{u,r'}^t)\right)}, \tag{21}$$

where $\mathbf{h}_{u,r}^t$ denotes the feature vector updated by neighbor under relation $r$ at time $t$, $\text{MLP}^t(\cdot)$ denotes time-wise multi layer perception at time $t$ with output dimension $d \in \mathbb{R}^1$.

### A.3.1 Study on Hierarchical Attention in HDGNNs

HDGNNs typically inherit a hierarchical attention mechanism from HGNNs to address the spatial heterogeneity present in HTGs. To analyze the effect of each level, the mean function is utilized to separately replace these two levels of attention. We conducted experiment on four real-world datasets, where Aminer and OGBN-MAG were used for link prediction, YELP was used for node classification, COVID-19 was used for node regression. We experiment on two representative HDGNN models, HTGNN [12] and DyHATR [11], where $\triangledown$ means removing the node-level attention and $\diamondsuit$ means removing the relation-level attention. Additionally, real-world datasets Aminer and OGBN-MAG are used for link prediction, while YELP is used for node classification.

As results shown in Table 7, the model without relation-level attention exhibited performance degradation, while the model without node-level attention did not. Additionally, we found that the computation time for node-level attention is approximately 5 times that of relation-level attention in DyHATR, with the former averaging 0.03 GPU seconds and the latter 0.006 seconds. This disparity is greater in HTGNN, where it is 8 times. Therefore, we obtain the first finding.

***Finding 1: Relation-level attention is vital, while node-level attention is trivial on HTGs.*** This may be attributed to the fact that intra-type neighbors in HTGs tend to exhibit lower variance compared to inter-type ones. This finding is also consistent with the latest research on HGNNs [29, 28, 30], which shows that a well-designed relation-level attention mechanism can be effective enough, even without node-level attention.

Table 7: Experiments to analyze the effects of hierarchical attentions. HTGNN [12] and DyHATR [11] are representative models for HTGs. $\triangledown$ means removing the node-level attention and $\diamondsuit$ means removing the relation-level attention.

| Dataset | Aminer | OGBN-MAG | YELP | COVID-19 |
|---|---|---|---|---|
| Metric | (AUC%) ↑ | (AUC%) ↑ | (Macro-F1%) ↑ | (MAE)↓ |
| DyHATR | **86.46 ± 1.28** | 89.49 ± 0.65 | 38.89 ± 0.64 | 668 ± 56 |
| DyHATR$^\triangledown$ | 86.37 ± 0.96 | **89.52 ± 0.49** | **39.16 ± 0.45** | **660 ± 39** |
| DyHATR$^\diamondsuit$ | 84.52 ± 1.48 | 87.49 ± 0.88 | 37.13 ± 0.84 | 698 ± 56 |
| HTGNN | 85.92 ± 0.93 | **91.21 ± 0.77** | 36.65 ± 1.13 | **555 ± 34** |
| HTGNN$^\triangledown$ | **85.95 ± 0.64** | 91.14 ± 0.95 | **36.87 ± 0.98** | 560 ± 28 |
| HTGNN$^\diamondsuit$ | 84.56 ± 0.82 | 89.45 ± 1.22 | 35.19 ± 1.43 | 601 ± 41 |

## B  Prompt Example

To show how we constructed the prompts for our model, a specific example of summary generation for the a node type (Academic papers) in the OGBN-MAG dataset is present in Figure 6. We generate distinct prompts for each type of node and input them into the large language model to obtain additional knowledge. We experimented with three LLMs: LLaMA3-8B, GPT-3.5 and LLaMA2-7B. As shown in Figure 8, SE-HTGNN enhanced by LLaMA3-8B achieves the best performance on four real-world HTG datasets.

Table 8: Comparison of different LLMs in performance.

| Dataset | Aminer | OGBN-MAG | YELP | COVID-19 |
|---|---|---|---|---|
| Metric | (AUC%) ↑ | (AUC%) ↑ | (Macro-F1%) ↑ | (MAE)↓ |
| SE-HTGNN$_{llama2-7B}$ | 89.92 ± 0.64 | 92.25 ± 1.08 | 42.84 ± 1.08 | 512 ± 12 |
| SE-HTGNN$_{GPT-3.5}$ | 90.53 ± 0.51 | 92.85 ± 0.98 | 43.37 ± 1.23 | 504 ± 8 |
| **SE-HTGNN$_{llama3-8B}$** | **91.08 ± 0.59** | **93.13 ± 0.56** | **44.24 ± 0.88** | **497 ± 5** |

Please note that in practice, we do not use the final text output of the LLMs. Instead, we take the embedding from the **last hidden layer** of the LLM as the output to facilitate subsequent computations. If the last hidden layer cannot be accessed (e.g., ChatGPT-3.5), we use an embedding model, such as text-embedding-ada-002[3], to convert the LLM's text output into embeddings for subsequent computations.

---

[3]https://platform.openai.com/docs/guides/embeddings

Figure 6: An example of constructing prompts for node type in OGBN-MAG Dataset.

# C  Experimental setup

## C.1  Dataset & Task Objective

We utilized two link prediction datasets: OGBN-MAG, Aminer, one node classification dataset, and one node regression dataset COVID-19. We follow the splits provided by previous works [12, 9]. For all tasks, we use the first $t$ snapshots for training, and the snapshots after $t + 1$ are used for validation and testing. The statistics of the datasets are summarized in Table 9, and their descriptions are as follows.

- **Aminer**[4]: Aminer [9] is an academic citation dataset for papers that were published during 1990-2006. The dataset has three types of nodes (paper, author and venue), and two types of relations (paper-*publish*-venue and author-*writer*-paper). **The task** is to predict links between author nodes, i.e., whether a pair of authors will coauthor a paper in the future.

- **OGBN-MAG**[5]: The original OGBN-MAG dataset is a static heterogeneous network composed of a subset of the Microsoft Academic Graph (MAG). HTGNN [12] extracts a heterogeneous temporal graph (HTG) from OGBN-MAG consisting of 10 graph snapshots spanning from 2010 to 2019. Specifically, previous work selects authors that consecutively publish at least one paper every year. Then it collects these authors' affiliated institutions, published papers, and the papers' field of studies in each year to construct this HTG. Each snapshots is a heterogeneous graph that contains four types of nodes (paper, author, institutions, and fields of study), and four types of relations among them (author-*affiliated with*-institution, author-*writes*-paper, paper-*cites*-paper, and paper-*has a topic of* -field of study). **The task** is to predict links between author nodes, i.e., whether a pair of authors will coauthor a paper in the future.

- **Yelp**[6]: Yelp [9] is a business review dataset, containing user reviews and tips on business. Following, we consider interactions of three categories of business including "American (New) Food", "Fast Food" and "Sushi" from January 2012 to December 2012. **The task** is to classify the type of business nodes, i.e., a three-class classification problem.

- **COVID-19**[7]: This dataset [12] contains both state and county level daily case reports (*e.g.*, confirmed cases, new cases, deaths, and recovered cases). We use the daily new COVID-19 cases as the time-series data for each state and county. We then build a HTG including 304 graph slices. Each graph slice is also a heterogeneous graph consisting of two types of nodes (state and county) and three types of relations between them, *i.e.*, one administrative affiliation relation (state-*includes*-county) and two geospatial relations (state-*near*-state, county-*near*-county). **The task** on COVID-19 is to predict the new daily cases.

---

[4]`https://www.aminer.cn/collaboration`
[5]`https://github.com/snap-stanford/ogb`
[6]`https://www.yelp.com/dataset`
[7]`https://coronavirus.1point3acres.com/en`

| Dataset | Graph | Time Span | Node | Relation | Data Split |
|---|---|---|---|---|---|
| Aminer | # Graph: 16
Granularity: year | 1990-2005 | # Paper : 18,464
# Author : 23,035
# Venue : 22 | # Paper-publish-Venue : 18,464
# Author-write-Paper : 52,545 | Training: 14
Validation: 1
Testing: 1 |
| OGBN-MAG | # Graph: 10
Granularity: year | 2010-2019 | # Author: 17,764
# Paper: 282,039
# Field: 34,601
# Institution: 2,276 | # Author-Paper: 2,061,677
# Paper-Paper:2,377,564
# Paper-Field: 289,376
# Author-Institution: 40,307 | Training: 8
Validation: 1
Testing: 1 |
| YELP | # Graph: 12
Granularity: month | 01/2012-
12/2021 | # User : 55,702
# Business : 12,524 | # User-review-Business : 87,846
# User-tip-Business : 35,508 | Training: 10
Validation: 1
Testing: 1 |
| COVID-19 | # Graph: 304
Granularity: day | 05/01/2020-
02/28/2021 | # State: 54
# County: 3223 | # State-State: 269
# State-County: 3,141
# County-County: 22,176 | Training: 244
Validation: 30
Testing: 30/60/90 |

Table 9: Statistics of datasets.

## C.2 Baseline

We compare our method with state-of-the-art baselines. Specifically, we compare SE-HTGNN with the following competitive baselines.

- **GCN** [46]: a representative static homogeneous GNN aggregating neighbors using degree normalized weights.

- **GAT** [26]: a representative static homogeneous GNN aggregating neighbors using the attention mechanism.

- **TGAT** [37]: a representative dynamic homogeneous GNN aggregating neighbors using the attention mechanism with temporal encoder.

- **RGCN** [17]: a static heterogeneous GNN that assigns different parameterizations for different relation types.

- **RGAT** [49]: a static heterogeneous GNN using the hierarchical attention mechanism that assigns different parameterizations for different node and relation types.

- **HGT** [8]: a static heterogeneous GNN adopting mutual attention and different attention parameterization for different node and relation types.

- **DiffMG** [19]: a representative static heterogeneous graph neural architecture search method. DiffMG automates the static heterogeneous GNN designs by searching meta-paths used by GCN and exploring the search space with its specially designed differentiable search algorithm.

- **DyHATR** [11]: a representative dynamic heterogeneous GNN that uses hierarchical attention and temporal self-attention to capture heterogeneous and temporal information.

- **HGT+** [8]: a dynamic heterogeneous GNN that extends HGT by utilizing the relative temporal encoding to model temporal information.

- **HTGNN** [12]: a dynamic heterogeneous GNN that uses hierarchical attention and temporal self-attention iteratively to capture complex dynamic heterogeneous information.

- **DHGAS** [9]: a representative dynamic heterogeneous graph neural architecture search method. DHGAS automates the dynamic heterogeneous GNN designs by searching attention method and exploring the search space with its specially designed multiple-stages differentiable search algorithm.

- **CasMLN** [13]: a novel attention-based heterogeneous dynamic GNNs that utilizes node degrees to calculate attention coefficients for that nodes. Furthermore, it utilize Large Language Model to provide auxiliary information from an alternative view.

Notably, among these methods, DiffMG and DHGAS are automated models, while the rest are manually designed models.

## C.3 Implementation Details and Hyperparameters

We use PyTorch-Geometric [52] implementations for GCN, GAT, RGCN, RGAT, HGT and HGT+. Other models, DiffMG[8], DyHATR[9], HTGNN[10], DHGAS[11], and CasMLN[12] are reproduced using the source code released by the authors. For all baselines and datasets, we use the default hyperparameters provided in the original source code, if available. Otherwise, we choose the number of message-passing layers in $\{1, 2, 3\}$ and the number of attention heads in $\{1, 2, 4, 8\}$. The hidden representation dimensionality is set as d = 64 except d = 8 for COVID-19. We record the best the best-performing results among them. Other hyperparameters for baselines are kept the same as in the original paper. The max number of epochs is 500, and we set the early stopping round on the validation set as 25 or 50 to alleviate over-fitting. We report the test performance based on the best epoch of the validation set. For our method, we adopt the Adam optimizer with a learning rate searched in $\{1,3,5\} \times \{10^{-2}, 10^{-3}\}$, and the weight decay rate is searched in $\{1,2,5\} \times \{10^{-4}, 10^{-5}\}$. The layer number of our methods is set as 2, except 3 for Aminer.

## C.4 Configurations

Experiments on all datasets are conducted with:

- Operating System: Ubuntu 20.04.6 LTS
- CPU: Intel(R) Xeon(R) CPU E5-2650 v2 @ 2.60GHz
- GPU: NVIDIA RTX 3090 with 24 GB of memory
- RAM: 128 GB
- Software: Python 3.9.19, Deep Graph Library[13] 1.1.1 [53], Cuda 11.3, PyTorch[14] 1.12.1[54], PyTorch-Geometric[15] 2.5.3 [52].

## C.5 Licenses

The licenses of the baselines and datasets are as follows:

- GNU Affero General Public License 3.0: COVID-19[16]
- MIT License: DyHATR[17], HGT[18], PyTorch-Geometric[19]
- Apache License 2.0: Yelp[20], Deep Graph Library [21]
- Other license: Unspecified license: Aminer, Ecomm, HTGNN, DiffMG
- Other license: PyTorch[22]

# D Limitations and Future Work

This work introduces large language models (LLMs) to enhance the semantic representations of node types, thereby improving the overall model performance. However, LLM-generated embeddings

---

[8]https://github.com/LARS-research/DiffMG

[9]https://github.com/skx300/DyHATR

[10]https://github.com/YesLab-Code/HTGNN

[11]https://github.com/wondergo2017/DHGAS/

[12]https://github.com/PasaLab/CasMLN

[13]https://www.dgl.ai/

[14]https://pytorch.org/

[15]https://github.com/pyg-team/pytorchgeometric

[16]https://coronavirus.1point3acres.com/en/data

[17]https://github.com/skx300/DyHATR/blob/master/LICENSE

[18]https://github.com/acbull/pyHGT/blob/master/LICENSE

[19]https://github.com/pyg-team/pytorch-geometric/blob/master

[20]https://www.yelp.com/dataset/

[21]https://github.com/dmlc/dgl/blob/master/LICENSE

[22]https://github.com/pytorch/pytorch/blob/master/LICENSE

typically have high and fixed dimensionality. As a result, the learnable transformation matrices required for linear dimensionality reduction contribute a considerable parameter overhead, which can compromise computational efficiency. Despite this, experimental results demonstrate that the incorporation of LLMs yields significant performance gains. In future work, we plan to explore dimensionality reduction techniques—such as low-rank decomposition and principal component analysis (PCA)—to compress LLM outputs, aiming to improve model efficiency and scalability without subsantially sacrificing semantic information.

## E  Broader Impacts

Our proposed SE-HTGNN is tailored neural network for Heterogeneous temporal graph (HTG), focusing on the efficient representation of large-scale graphs problem. The application of the COVID-19 dataset shows that heterogeneous temporal graph (HTG) learning models such as SE-HTGNN can effectively predict the development trend of the epidemic, which is helpful for the government to intervene in advance, optimize the allocation of medical resources, and enhance the social emergency response capacity. On the other hand, in e-commerce networks YELP, HTG learning models like SE-HTGNN can greatly enhance user experience by capturing the dynamic interactions between users, items, and contextual factors over time, enabling more accurate personalized recommendations, trend forecasting, and fraud detection.

