# OpenReview forum: "Simple and Efficient Heterogeneous Temporal Graph Neural Network"
_NeurIPS.cc/2025/Conference — NeurIPS 2025 poster_

### Official Review · Reviewer_YMqu · 2025-06-24

**Clarity:** 4
**Significance:** 3
**Originality:** 3
**Rating:** 5
**Confidence:** 4

**Summary:**

This work proposes SE-HTGNN, a simplified heterogeneous temporal graph neural network (HTGNN) or heterogeneous temporal graph (HTG). It replaces complex attention mechanism of HTGNNs in recent literature by simple GRU outputs overtime, and replaces complex message aggregation by non-parametric aggregation. It achieves state-of-the-art performance on various benchmarks with far better efficiency. Besides, this work also utilizes LLM for initialization to improve the performance taking advantage from pretrained models.

**Questions:**

- line 123: Shouldn't be |T_n| + |Y_r| > 2? Since |T_n| >= 1, |T_r| >= 1 and either should be more than one to have heterogeneity.
- sec 4.2: I think relation should also be considered in the attention construction. Do you omit it just for efficiency? Or is you implicitly assume that there will only be one relation type between arbitarily two node types?
- Eq (8): Can you explain how multi-head version it will be? Or you are just use one-head in your experiment.
- Ablation study: Can you explain what is used as replacement in w/o att_1, a gated attention or a self-attention mechanism? convention attention can refer many cases.
- Table 4: What is d'?
- Visualization (2): It seems to show an attention change between two snapshots, but I can not get any valuable conclusion from this paragraph. Besides, is it a simplied example? The attention sum to one, but it is hard to image a state to have one neighbor state or one county.

**Ethical Concerns:**

["NO or VERY MINOR ethics concerns only"]

**Final Justification:**

Most notations, typo concerns are addressed. Except the expressivity insufficency which slightly limit the significance of the work, thus will keep accept score.

**Limitations:**

Yes.

Also some missing limitations are introduced in weakness.

**Paper Formatting Concerns:**

No concern.

**Quality:**

4

**Strengths And Weaknesses:**

## Strength

- This work uses simple architecture to achieve better performance, which is a good innovation for practical usage on large graphs given the literature always focus on more complex achitecture.
- This work can achieve SOTA or same performance on various benchmarks with far less cost (without using LLM initialization).
- The experiment results are sound, giving evidences from various aspect to justify the improvement of proposed method.

## Weakness

- This work only compares on four benchmarks, I think more benchmarks will make the experimental result more trustworthy.
- This work only consider snapshot temporal graph, however, I believe event (continuous) temporal graph is more common in the literature of HTG, this limits the significance and impact of this work.
- This work is still fall into aggregate-graph-first aggregate-time-second framwork, which has been theoretically proved to have flawness in compare to TGAT or TGN framework [1], thus limit the power of this work (but is still a good work).

[1] Gao, Jianfei, and Bruno Ribeiro. "On the equivalence between temporal and static equivariant graph representations." International Conference on Machine Learning. PMLR, 2022.

---

> ### Author Rebuttal · Authors · 2025-07-31
>
> Dear Reviewer YMqu, we are deeply grateful for your careful reading and positive feedback on the motivation and contributions of our work. We hope to have carefully addressed your concerns and made updates to the manuscript based on your feedback.
> Here is our response to your comments and questions.
>
> ---
>
> ###  **Question 1**：Shouldn't line 123 be $|T_n| + |Y_r|$ > 2?
>
> We fully agree with your comment that having $|T_n| > 1$ or $|T_r| > 1$ is essential to reflect true heterogeneity in a graph. Our choice of the condition  $|T_n|+|T_r| ≥ 2$ was motivated by the following considerations:
> - Prior works [1,2] often adopt a $|T_n|+|T_r| ≥ 2$ formulation to define HTGs.
> - Many existing graph settings—such as homogeneous temporal graphs, heterogeneous static graphs, and others—can be regarded as special cases of HTGs under this broader formulation.
> We have clarified this definition in the revised manuscript to prevent further confusion.
>
> [1] Zhang Z, et al. "Dynamic Heterogeneous Graph Attention Neural Architecture Search" AAAI 2023.
> [2] Wang F, et al. "LLM-enhanced Cascaded Multi-level Learning on Temporal Heterogeneous Graphs" ACM SIGIR 2024.
>
> ---
>
> ###  **Question 2**: Relation should also be considered in the attention construction. Do you omit it just for efficiency? More benchmark will make the experimental result more trustworthy.
>
> Thank you for your thoughtful suggestion! We would like to clarify that relation types are not omitted in our attention construction.
> We provide explanations from the perspectives of both the *Dynamic Attention* module and the *LLM-Enhanced Prompt* module:
>
> - **For Dynamic Attention**, as shown in our formulation of the dynamic attention (detailed in line 183):
> $e^{t-1}_r=GRU_r (H_r, e^{t-1}_r), $
> we employ relation-wise $GRU_r(\cdot)$to independently model the attention coefficients for different relation types. Particularly, if multiple relations exist between two node types, we also apply an independent $GRU(\cdot)$ for each relation to compute the respective attention coefficients. In fact, such multi-relation scenarios are present in our **Yelp dataset**.
>
> - **For LLM-Enhanced Prompt**,  the node-type prompts inherently clarify the relation type. Particularly, in multi-relational graphs, for each source-target node pair connected via different relations, we create separate node-type prompt variants, each explicitly specifying the corresponding relation. This design prevents different relations connecting the same node types from sharing the same initial coefficient.
>
> - **Additional Multi-relation Benchmark.** To further validate performance on multi-relation graph,  we introduce a new multi-relational benchmark called **Ecomm** [3] from the Alibaba Tianchi competition, where nodes type (e.g., *user* and *item*) can be connected through various relations such as *click*, *like*, *cart* or *buy*.
> The newly introduced dataset also involves a link prediction task, and the experimental results are presented as follows.
> |  *Ecomm*     |     GAT     |     TGAT    |     RGCN    |     RGAT     |    DiffMG   |   DyHATR   |     HGT+    |    HTGNN   |    DHGAS   |   CasMLN   | **Ours** |
> |:------:|:-----------:|:-----------:|:-----------:|:-----------:|:-----------:|:----------:|:-----------:|:----------:|:----------:|:----------:|:----------:|
> | AUC(%)  | 56.61±1.45  | 67.19±0.98  | 68.65±0.48  | 68.84±0.86    | 60.49±0.89  | **69.62±0.50** |  61.46±2.56 | 58.58±0.63 | 62.56±0.58 | 67.68±0.47 | ***70.96±0.43***|
> |  AP(%) |  56.88±1.31 |  69.78±0.64 | 71.49±0.42  |  71.57±0.73  | 62.83±0.67  | **73.40±0.50** | 63.79±1.83  | 60.74±0.82 | 63.74±0.82 | 68.72±0.41 | ***74.15±0.42*** |
> |                     |             |             |             |             |                         |            |             |            |            |            |            |
>
> Our model achieves sota performance on this dataset as well, demonstrating its ability to effectively handle multi-relational heterogeneous graphs.
> We will include these experimental results and clarification in the main text or appendix. Once again, we sincerely appreciate your comment, which helped improve the clarity of the manuscript.
>
> [3] Xue H, et al. "Modeling dynamic heterogeneous network for link prediction using hierarchical attention with temporal rnn" ECML 2020.
>
> ---
>
> ###  **Question 3**: Can you explain how multi-head version of Eq(8) it will be?
>
> Thank you for the question. Our model’s multi-head version can be implemented by adjusting the hidden state dimension of the GRU. As shown in the equation, $e^{t-1}\_{r}=GRU\_{r} (H\_{r}, e^{t-1}\_{r})$ ,
> where $e^{t}\_{r} \in R^{n \times k}$ denotes the hidden state, with $n$ being the number of nodes and $k$ being the number of attention heads. We take the average value across the k heads (i.e.,  $mean(e^{t}_{r})$), to obtain the final attention coefficient.
>
> Furthermore, we conducted additional experiments to evaluate the effect of varying heads.
> | multi-head | OGBN_MAG |  | AMiner |  | YELP |  | COVID19 |  |
> |---|---|---|---|---|---|---|---|---|
> |  | AUC | AP | AUC | AP | MacroF1 | Recall | MSE | RMSE |
> | 1 | 93.13±0.56  | 92.71±0.52  | 91.08±0.59  | 90.03±0.48 | 44.24±0.88  | 44.68±0.43 | 497±5 | 1069±11 |
> | 2 | 93.08±0.42 | 92.68±0.40  | 91.15±0.48 | 90.09±0.39 | 43.98±0.94 | 44.36±0.94 | 493±7 | 1058±14 |
> | 4 | 93.22±0.32 | 92.94±0.26 | 90.97±0.21 | 90.14±0.18 | 44.36±0.74 | 44.71±0.44 | **488±8** | **1047±12** |
> | 8 | **93.31±0.35** | **93.01±0.22** | **91.31±0.28** |**90.28±0.20** | **44.42±0.69** | **44.82±0.38** | 490±4 | 1052±9 |
> |  |  |  |  |  |  |  |  |  |
>
>  As shown in the results, increasing the number of heads leads to more stable performance and moderate improvements, as the higher hidden dimension allows the model to encode richer historical information.
> To balance efficiency and effectiveness, we adopt head = 1, which already yields competitive performance with minimal computational cost. These results and analyses have been included in the revised manuscript.
>
> ---
>
> ###  **Question 4**: Can you explain what is used as replacement in w/o $ att_1$?
>
> In Table 3, the alternative attention mechanism is a simple projection-based attention (detailed see Line 530, it seems that the text box cannot properly render our equation), which has been widely adopted in prior work [2-4]. For each relation $r$, the attention coefficients $\beta_r$ are computed as follows:
>
> $\beta_r = \frac{\exp\left(MLP(H_{u,r})\right)}{\sum_{r' \in R} \exp\left(MLP(H_{u,r\prime}\right))},$
>
> where $H_{u,r}$ denotes the representation  updated by neighbor under relation $r$, MLP($\cdot$) denotes multi layer perception.
> In addition, we conducted ablation studies comparing gate-attention and self-attention variants, with results shown below.
>
> | Ablation | OGBN_MAG |  | AMiner |  | YELP |  | COVID19 |  |
> |---|---|---|---|---|---|---|---|---|
> |  | AUC | AP | AUC | AP | MacroF1 | Recall | MSE | RMSE |
> | Self-attention | 91.65±1.22 | 90.65±1.14 | 88.73±0.82 | 88.21±0.94 | 42.41±1.32 | 42.73±0.97 | 545±33 | 1114±42 |
> | Gate-attention | 87.94±2.42 | 87.24±1.89 | 87.42±1.24 | 86.55±1.33 | 38.96±3.27 | 39.26±2.27 | 574±45 | 1216±68 |
> | Original | **93.13±0.56**  | **92.71±0.52**  | **91.08±0.59**  | **90.03±0.48** | **44.24±0.88**  | **44.68±0.43** | **497±5** | **1069±11** |
> |  |  |  |  |  |  |  |  |  |
>
> From the results, both attention variants outperform the projection-based attention. However, their overall performance is still inferior to our original model. These findings have been added to the revised manuscript.
>
> [4] Wang X, et al. "Heterogeneous graph attention network" WWW 2019.
>
> ---
>
> ###  **Question 5**: What is d' in Table 4?
>
> Here,  d' denotes the hidden dimension of the large language model. Since this value is typically much smaller than the number of edges (i.e.,  d'<< |e|), it does not contribute significantly to the overall time complexity. We have added this clarification to the revised manuscript.
>
> ---
>
> ###  **Question 6**: More detailed description of Visualization (2).
>
> Thank you for your comment. Visualization (2) illustrates how attention coefficients evolving over a 30-day period for a specific U.S. state in the COVID-19 dataset. In this figure:
> - The blue "state" denotes a specific state node.
> - The orange "state" refers to **all** neighboring state-type nodes of the blue state.
> - The "county" refers to **all** county-type nodes governed by the blue state.
> - The numbers reflect the trend of attention coefficients changes over time.
>
> In the task of predicting the new cases of COVID-19 in that state, we observe that the attention coefficients assigned to neighboring states are consistently higher than those for county. This suggests that the state's case increases are likely more influenced by adjacent states.
> Furthermore, the coefficients of the state-type node increases from 0.58 to 0.61, indicating that recent information from the neighboring state becomes more important than historical data.
> This observation also supports our insight that historical attention patterns are useful for future learning. Specifically, in this case, we see that state-type deemed important in historical snapshot retain or even increase their importance later on.
> We have included this interpretation in the revised manuscript and updated the visualization to improve clarity and readability.

---

> ### Author Response · Authors · 2025-08-06
>
> Dear Reviewer YMqu,
> we sincerely appreciate your supportive feedback and constructive comments, which have helped us improve the manuscript. Should there be any remaining points where we can provide further clarification, feel free to let us know.

---

### Official Review · Reviewer_KYMR · 2025-07-02

**Clarity:** 3
**Significance:** 2
**Originality:** 3
**Rating:** 4
**Confidence:** 4

**Summary:**

This paper introduces a novel and efficient framework for learning on heterogeneous temporal graphs, called SE-HTGNN, integrating temporal information directly into spatial learning using a dynamic attention mechanism guided by GRU. Experiments conducted on four real-world datasets demonstrate that SE-HTGNN achieves state-of-the-art performance while offering significantly faster training speeds and reduced model complexity.

**Questions:**

See the weakness part.

**Ethical Concerns:**

["NO or VERY MINOR ethics concerns only"]

**Final Justification:**

The authors' response has fully addressed my concerns. Therefore, I have decided to maintain my positive score.

**Limitations:**

Yes.

**Paper Formatting Concerns:**

None.

**Quality:**

2

**Strengths And Weaknesses:**

Strengths:
- Semantic attention initialiation. The model leverages LLMs to generate semantic embeddings for node types, utilizing these embeddings to initialize attention scores, enhancing generalization and improves type-awareness within the graph learning process.
- Efficient spatial and temporal integration. SE-HTGNN proposes to decouple spatial and temporal modeling by integrating temporal learning directly into the graph learning process, improving both efficiency and representation capability.
- Simplified representation. By eliminating redundant hierarchical attention and temporal modules, the model adopts a simplified aggregation and projection approach to reduce computational complexity and resource consumption.

Weaknesses:
- Additional datasets. TGB 2.0 is recommended to further validate the model's performance across diverse settings.
- Explanation of Table 3. The ablation studies of Table 3 presents two attention variants regading the specific components, requiring more detailed explanation would improve transparency and reproducibility.
- Analysis on the GRU component. The GRU component should be compared to alternative components to highlight its necessity and effectiveness. The random initialization of GRU hidden states should also be analyzed to verify the effectiveness of the proposed initialization method.

---

> ### Author Rebuttal · Authors · 2025-07-31
>
> Dear Reviewer KYMR, we are deeply grateful for your careful reading and positive feedback of our work. We hope to have carefully addressed your concerns and made updates to the manuscript based on your feedback. Here is our response to your comments and questions.
>
> ---
> ###  **Question 1**: Additional datasets. TGB 2.0 is recommended.
>
> Thank you for pointing out this valuable benchmark! In the revised version of the paper, we have cited the TGB 2.0 [1] references and added a detailed discussion in the Related Work section.
>
> However, upon close examination of the TGB 2.0, we found that they are constructed as **continuous-time** heterogeneous graphs, whereas our method is specifically designed for **discrete-time** (snapshot-based) graphs. As highlighted in a survey  of dynamic graph [2,3],  these two temporal paradigms correspond to different application scenarios and present distinct modeling challenges and evaluation strategies. The divergence between the baselines adopted in TGB 2.0 and those used in our work also shows the difference in problem settings. We are regrettably unable to incorporate this benchmark into our evaluation.
>
> To further validate the model's performance, we introduce a new real-world discrete-time benchmark called **Ecomm** [4] from the Alibaba Tianchi competition processed by previous work, where nodes type (e.g., *user* and *item*) can be connected through various relations such as *click*, *like*, *cart* or *buy*.
> The newly introduced dataset also involves a link prediction task (recommend item for user) , and the experimental results are presented as follows.
>
> |  *Ecomm*     |     GAT     |     TGAT    |     RGCN    |     RGAT     |    DiffMG   |   DyHATR   |     HGT+    |    HTGNN   |    DHGAS   |   CasMLN   | **Ours** |
> |:------:|:-----------:|:-----------:|:-----------:|:-----------:|:-----------:|:----------:|:-----------:|:----------:|:----------:|:----------:|:----------:|
> | AUC(%)  | 56.61±1.45  | 67.19±0.98  | 68.65±0.48  | 68.84±0.86    | 60.49±0.89  | **69.62±0.50** |  61.46±2.56 | 58.58±0.63 | 62.56±0.58 | 67.68±0.47 | ***70.96±0.43***|
> |  AP(%) |  56.88±1.31 |  69.78±0.64 | 71.49±0.42  |  71.57±0.73  | 62.83±0.67  | **73.40±0.50** | 63.79±1.83  | 60.74±0.82 | 63.74±0.82 | 68.72±0.41 | ***74.15±0.42*** |
> |    |   | | |   |     |   |    |   |    |     |      |
>
> Our model achieves sota performance on this dataset as well, demonstrating its ability to effectively handle multi-relational heterogeneous graphs. The detailed description of the datasets and experimental results has been added to the revised manuscript.
>
>
> [1] Gastinger J, et al. Tgb 2.0: A benchmark for learning on temporal knowledge graphs and heterogeneous graphs. Nips 2024.
> [2] Barros C D T, et al. "A survey on embedding dynamic graphs" ACM Computing Surveys 2021.
> [3] Kazemi S M,  et al. "Representation learning for dynamic graphs: A survey" Journal of Machine Learning Research 2020.
> [4] Xue H, et al. "Modeling dynamic heterogeneous network for link prediction using hierarchical attention with temporal rnn" ECML 2020.
>
> ---
>
> ###  **Question 2**: More Explanation of Table 3.
>
> Thank you for your suggestions. Specifically, we implemented two variants to replace the dynamic attention module: w/o $ att_1$ and w/o $ att_1$.
> - For **w/o $ att_1$**, the alternative attention mechanism is a simple projection-based attention (detailed see Line 530, it seems that the text box may cannot properly render our equation), which has been widely adopted in prior work [4-6]. For each relation $r$, the attention coefficients $\beta_r$ are computed as follows:
> $\beta_r = \frac{\exp\left(MLP(H_{u,r})\right)}{\sum_{r' \in R} \exp\left(MLP(H_{u,r\prime}\right))},$
> where $H_{u,r}$ denotes the representation updated by neighbor under relation $r$, MLP($\cdot$) denotes multi layer perception.
>
> - For **w/o $ att_2$**, we completely remove the attention mechanism, meaning that representations from all types are aggregated by simple averaging.  Specifically, the process is formulated as:
> $H_{v} = \frac{1}{|\mathcal{R}(v)|} \sum_{r \in \mathcal{R}(v)}  H_{v,r}$,
> where $|\mathcal{R}(v)|$ denotes the number of relation types.
>
> We have incorporated these detailed descriptions into the revised manuscript.
>
> [5] Wang X, et al. "Heterogeneous graph attention network" WWW 2019.
> [6] Wang F, et al. "LLM-enhanced Cascaded Multi-level Learning on Temporal Heterogeneous Graphs" ACM SIGIR 2024.
>
> ---
>
> ###  **Question 3**: More analysis on the GRU component with alternative components, the random initialization of GRU hidden states should also be analyzed.
>
> Thank you very much for the constructive suggestion! We have conducted additional ablation studies on the GRU module. The results are summarized below.
>
> -  **Alternative Components.** We add comparative experiments for the GRU with its alternative components as follows: **LSTM** [7], **Transformer** [8], and **Mamba** [9]. Specifically, both LSTM and Mamba are RNN-like models and can directly replace the GRU module in our framework.  For the Transformer, we first apply self-attention over the historical representations to aggregate information across all time steps, and then project the output into the attention coefficient space.
> It is worth noting that the Transformer does not maintain a hidden state over time, and therefore cannot benefit from the LLM-enhanced prompt module. The results are summarized below.
> | Alternative  | OGBN_MAG |  | AMiner |  | YELP |  | COVID19 |  |
> |---|---|---|---|---|---|---|---|---|
> |  | AUC | AP | AUC | AP | MacroF1 | Recall | MSE | RMSE |
> | LSTM | 92.77±0.48 | 92.28±0.45 | 90.52±0.48 | 89.87±0.58 | **44.31±0.97** | **44.73±0.52** | 506±7 | 1078±16 |
> | Mamba | 91.79±1.32 | 90.92±1.45 | 87.81±1.65 | 87.12±1.59 | 41.45±2.41 | 41.58±3.10 | 601±67 | 1259±84 |
> | Transformer | 90.75±0.32 | 89.93±0.34 | 90.93±0.45 | 89.89±0.32 | 43.59±1.21 | 43.64±1.19 | 529±12 | 1098±23 |
> | Original (GRU) | **93.13±0.56**  | **92.71±0.52**  | **91.08±0.59**  | **90.03±0.48**| 44.24±0.88  | 44.68±0.43 | **497±5** | **1069±11** |
> |  |  |  |  |  |  |  |  |  |
>
> According to the experimental results, the original model (GRU-based) and the LSTM-based variant achieve comparable performance. However, GRU, as a simplified version of LSTM, offers better computational efficiency.  In contrast, both Transformer-based and Mamba-based variant underperform in this task. For Mamba model, as reported in its original paper,  it suffers from unstable training, which may account for its inferior performance. For Transformer, one possible reason is that the Transformer can't benefit from the LLM-enhanced prompt.  We have incorporated these results, implement detail and analyses into the revised manuscript.
>
> -  **Initialization of GRU.** We evaluated three initialization strategies as follows:
>     -  **Random**: Initialized with values randomly sampled from a standard normal distribution.
>     -  **Average**: Initialized with uniform values based on the number of relation types (n), i.e., each relation is assigned a value of 1/n.
>     -  **Zero**: All initial values are set to zero.
> The corresponding experimental results are summarized below.
> | Initialization  | OGBN_MAG |  | AMiner |  | YELP |  | COVID19 |  |
> |---|---|---|---|---|---|---|---|---|
> |  | AUC | AP | AUC | AP | MacroF1 | Recall | MSE | RMSE |
> | Random | 90.87±1.24  | 90.06±1.27  | 87.91±1.54 | 87.05±189 | 41.05±0.93 | 41.29±0.68 | 542±28 | 1181+46 |
> | Average | 91.18±0.81 | 89.83±1.29 | 89.76±0.34 | 88.56±0.33 | 43.39±0.62 | 43.82±0.78 | 521±22 | 1102+35 |
> | Zero | 91.78±0.68  | 91.08±0.65  | 89.98±0.62  | 88.93±0.76 | 43.31±0.79  | 43.76±1.07 | 524±9 | 1114±19 |
> | Original | **93.13±0.56**  |  **92.71±0.52**  |  **91.08±0.59** |  **90.03±0.48** |  **44.24±0.88** |  **44.68±0.43** |  **497±5** |  **1069±11** |
> |  |  |  |  |  |  |  |  |  | |
>
> According to the experimental results, the random initialization performs the worst, as it essentially introduces noise into the model. The Zero initialization performs slightly better, since it does not carry any prior positive or negative bias. However, both approaches are outperformed by our original initialization strategy, which leverages LLM-generated prompts to provide semantically meaningful initial values.
> We have incorporated these results and their corresponding analysis into the revised manuscript.
>
> [7] Hochreiter S and Schmidhuber J. "Long short-term memory". Neural computation 1997.
> [8] Vaswani A, et al. "Attention is all you need". NIPS 2017.
> [9] Gu A, Dao T. "Mamba: Linear-time sequence modeling with selective state spaces". arXiv 2023.

---

> > ### Comment · Reviewer_KYMR · 2025-08-03
> >
> > The authors' response has fully addressed my concerns. Therefore, I have decided to maintain my positive score.

---

> ### Author Response · Authors · 2025-08-04
>
> We truly appreciate the reviewer's feedback and their decision to maintain the initial positive score. Thank you for your time and valuable comments.

---

### Official Review · Reviewer_FjxQ · 2025-07-02

**Clarity:** 3
**Significance:** 2
**Originality:** 2
**Rating:** 4
**Confidence:** 4

**Summary:**

This paper proposes SE-HTGNN, a new paradigm that integrates temporal modeling into spatial learning through a dynamic attention mechanism, which retains historical attention coefficients to guide future attention computations. Additionally, Large Language Models (LLMs) are introduced to initialize attention mechanisms via type-specific prompts, injecting external semantic knowledge. Experiments on four real-world datasets show SE-HTGNN achieves significant improvements in both performance and efficiency.

**Questions:**

1.	Could you provide experimental evidence demonstrating that current HDGNNs exhibit high model complexity?

2.	Could you provide the papers that stack additional attention layers and assign non-shared parameters for each snapshot?

3.	Could you provide an ablation study for the Simplified Neighbor Aggregation technique?

4.  Does storing historical attention coefficients in the hidden state of GRU limit the expansion of SE-HTGNN to large-scale Heterogeneous Temporal Graphs?

5. Can the Dynamic-Attention-based Fusion designed in this paper be replaced in other sequence-based modules such as the recurrent neural network (RNN) or Transformer?

**Ethical Concerns:**

["NO or VERY MINOR ethics concerns only"]

**Final Justification:**

After checking the results provided in response, all my concerns are addressed.

**Limitations:**

Yes

**Quality:**

2

**Strengths And Weaknesses:**

**Strengths**

S1.	SE-HTGNN achieves up to 10× speed-up over the latest baselines while maintaining well forecasting accuracy.

S2.	The learning paradigm for HTGs designed in this paper incorporates historical attention information, which facilitates capturing consistent long-term patterns.

S3.	The paper utilizes LLMs to generate semantic representations of node types, which enhances the model's domain understanding capability.

**Weaknesses**

W1.	The definition of Heterogeneous Temporal Graph is problematic. Since G^t already contains V^t, E^t, and X^t, the formula \mathcal{G} = \left( \{ G^t \}_{t=1}^T, \mathcal{V}, \mathcal{E}, \mathcal{X} \right) constitutes redundant definition. You can refer to the CasMLN paper[1] to revise the definition.

W2.	The LLM-enhanced prompt, proposed as one of the contributions in this paper, exhibits limited novelty in comparison to existing LLM-based techniques.

W3.	The technical contributions of this paper are insufficient, and it lacks the theoretical analysis.

>[1]	Wang F, Zhu G, Yuan C, et al. LLM-enhanced Cascaded Multi-level Learning on Temporal Heterogeneous Graphs[C]//Proceedings of the 47th International ACM SIGIR Conference on Research and Development in Information Retrieval. 2024: 512-521.

---

> ### Author Rebuttal · Authors · 2025-07-31
>
> Dear Reviewer FjxQ, we are deeply grateful for your careful reading and valuable feedback of our work. We hope to have carefully addressed your concerns and made updates to the manuscript based on your feedback. Here is our response to your comments and questions.
>
> ###  **Question 1**:  experimental evidence demonstrating high model complexity of current HDGNNs.
>  We have reported both the theoretical **time complexity** and the actual training time in Table 4 and Table 5, respectively (see Line 317). In addition, we further compare the GPU memory usage (in MB)of different methods as the time_window increases, using two challenging settings: a large-scale graph with millions of edges (OGBN-MAG) and a long-sequence dataset (Covid19). The detailed results are summarized below.
> | OGBN_MAG (Time_window): | 2 | 3 | 4 | 5 | 6 | 7 |
> |:---:|:---:|:---:|:---:|:---:|:---:|:---:|
> | HTGNN | 2622  | 5249 | 8719 | 13084 | 18131 | OOM |
> | DyHART | 3648 | 7266 | 12067 | 18091 | OOM | OOM |
> | CasMLN | 5058 | 6540 | 8014 | 9587 | 11038 | 12597 |
> | Ours | **1134** | **2110** |**3440**|**5094**|**6585**|**8036**|
> ||||||||
>
> | Covid19 (Time_window): | 3 | 7 | 14 | 21 | 28 | 60 |
> |:---:|:---:|:---:|:---:|:---:|:---:|:---:|
> | HTGNN | 24 | 58 | 119 | 185 | 255 | 602 |
> | DyHART | 40 | 82 | 159 | 238 | 361 | 856 |
> | CasMLN | 86 | 157 | 196 | 210 | 235 | 354 |
> | Ours | **24** | **48**|**86**|**114**|**149**|**238**|
> ||||||||
>
> The experimental results show that our method consistently maintains the lowest memory usage. In contrast, DyHART and HTGNN encounter out-of-memory (OOM) issues on large graphs, which demonstrating the high model complexity.
>
> ---
>
> ###  **Question 2**:  provide the papers that stack additional attention layers and assign non-shared parameters for each snapshot?
> Thank you for your questions. We summarize the relevant works in chronological order: DHAN [1] extends DySAT [2] from dynamic homogeneous graphs to dynamic heterogeneous graphs by stacking an additional edge-level attention mechanism layer. Then, DyHART [3] builds upon DHAN by stacking an additional GRU layer. HTGNN [4] improves on DyHART by assigning non-shared parameters for each snapshot. CasMLN [5] extends their approach by introducing LLMs into the framework.
> We have revised the Related Work section of the manuscript and better clarify the technical lineage and innovations.
>
> [1] Yang L, et al. "Dynamic heterogeneous graph embedding using hierarchical attentions" ECIR 2020.
> [2] Sankar A, et al. "Dysat: Deep neural representation learning on dynamic graphs via self-attention networks" WSDM 2020.
> [3] Xue H, et al. "Modeling dynamic heterogeneous network for link prediction using hierarchical attention with temporal rnn" ECML 2020.
> [4] Fan Y, et al. "Heterogeneous temporal graph neural network"  SDM 2022.
> [5] Wang F, et al. "LLM-enhanced Cascaded Multi-level Learning on Temporal Heterogeneous Graphs" ACM SIGIR 2024.
>
> ---
>
> ###  **Question 3**:  provide an ablation study for the Simplified Neighbor Aggregation.
> Thank you for your constructive comments. We have provided additional ablation studies on the Simplified Neighbor Aggregation module. Specifically, we propose four variants:
> - Non-aggregation.
> - GCN with shared parameters across snapshots.
> - GCN with independent parameters.
> - GAT with independent parameters.
> - GAT with shared parameters.
>
> The experimental results are summarized in the figure below.
>
> | |OGBN_MAG| |AMiner|  |YELP|  |COVID19| |
> |---|---|---|---|---|---|---|---|---|
> |  |AUC|AP|AUC|AP|MacroF1|Recall|MSE|RMSE|
> |Non-aggregation|83.91±1.20 | 82.60±1.16|62.47±1.23|64.91±1.18|35.27±2.79|35.56±2.52|672±78|1336±131|
> |GCN (Independent) |90.65±0.93 | 87.24±1.76 | 89.19±0.28 | 88.79±0.24| 44.19±0.68 | 44.21±0.52 |502±13|1074±26|
> |GCN (Shared) |90.57±0.86 | 92.28±0.75  | 88.40±2.64 | 87.26±2.21 | 43.69±0.79  | 43.82±0.62|508±10|1080±18|
> |GAT (Independent) | **93.24±1.28** |92.46±0.82|90.37±1.24|89.69±1.31|43.21±0.88| 43.59±0.79|566±74 |1219±140|
> |GAT (Shared)|91.93±0.46|89.54±0.48|88.37±0.28|86.89±0.61|42.11±1.32|42.23±1.41|605±88|1231±137|
> |SE-HTGNN |93.13±0.56|**92.71±0.52**|**91.08±0.59**|**90.03±0.48**|**44.24±0.88**|**44.68±0.43**|**497±5**|**1069±11**|
> | |  |  |  |  |  |  |  |  |
>
> We summarize our observations as follows:
> (1) Non-aggregation unsurprisingly achieves the lowest performance, as it effectively removes the core graph learning component.
> (2) The GAT and GCN (Independent) variants demonstrate competitive performance on certain datasets. However, this comes at a significant computational cost.
> (3) Our original method consistently maintains superior performance across most datasets, demonstrating a good balance between effectiveness and efficiency.
> The experimental results and implementation details have been incorporated into the revised manuscript.
>
> ---
>
> ###  **Question 4**:  Does storing historical attention coefficient limit the expansion?
>
> Thank you for your feedback. As shown in the tables provided in our response to **Question 1**, our method consistently maintains the lowest memory consumption under both large-scale graphs (with millions of edges) and long time-window settings.
>  This demonstrates that storing historical attention coefficients does not hinder the scalability of our approach.
>
> ---
>
> ###  **Question 5**:  Can the Dynamic-Attention-based Fusion designed in this paper be replaced?
>
> Thank you for your constructive suggestions, we address the reviewer’s concerns from two experiments: (1) by replacing on GRU with other sequence-based modules, and (2) by replacing the overall attention mechanism.
>
> - **For the replacement of GRU**,  we replace the GRU with other sequence-based modules as follows: **LSTM** [6], **Transformer** [7], and **Mamba** [8]. It is worth noting that the Transformer does not maintain a hidden state over time, and therefore cannot benefit from the LLM-enhanced prompt module. The results are summarized below.
> | Alternative  | OGBN_MAG |  | AMiner |  | YELP |  | COVID19 |  |
> |---|---|---|---|---|---|---|---|---|
> |  | AUC | AP | AUC | AP | MacroF1 | Recall | MSE | RMSE |
> | LSTM | 92.77±0.48 | 92.28±0.45 | 90.52±0.48 | 89.87±0.58 | **44.31±0.97** | **44.73±0.52** | 506±7 | 1078±16 |
> | Mamba | 91.79±1.32 | 90.92±1.45 | 87.81±1.65 | 87.12±1.59 | 41.45±2.41 | 41.58±3.10 | 601±67 | 1259±84 |
> | Transformer | 90.75±0.32 | 89.93±0.34 | 90.93±0.45 | 89.89±0.32 | 43.59±1.21 | 43.64±1.19 | 529±12 | 1098±23 |
> | Original (GRU) | **93.13±0.56**  | **92.71±0.52**  | **91.08±0.59**  | **90.03±0.48**| 44.24±0.88  | 44.68±0.43 | **497±5** | **1069±11** |
> |  |  |  |  |  |  |  |  |  |
>
> According to the experimental results, the original model (GRU-based) and the LSTM-based variant achieve comparable performance. However, GRU, as a simplified version of LSTM, offers better computational efficiency.  In contrast, both Transformer-based and Mamba-based variant underperform in this task. For Mamba model, as reported in its original paper,  it suffers from unstable training, which may account for its inferior performance.
>
> - **For the replacement of entire attention,** we replaced the original dynamic attention-based fusion module with self-attention and gate-attention. The experimental results are summarized as follows.
> | Ablation | OGBN_MAG |  | AMiner |  | YELP |  | COVID19 |  |
> |---|---|---|---|---|---|---|---|---|
> |  | AUC | AP | AUC | AP | MacroF1 | Recall | MSE | RMSE |
> | Self-attention | 91.65±1.22 | 90.65±1.14 | 88.73±0.82 | 88.21±0.94 | 42.41±1.32 | 42.73±0.97 | 545±33 | 1114±42 |
> | Gate-attention | 87.94±2.42 | 87.24±1.89 | 87.42±1.24 | 86.55±1.33 | 38.96±3.27 | 39.26±2.27 | 574±45 | 1216±68 |
> | Original | **93.13±0.56**  | **92.71±0.52**  | **91.08±0.59**  | **90.03±0.48** | **44.24±0.88**  | **44.68±0.43** | **497±5** | **1069±11** |
> ||||||||||
>
> From the results, the performance of variants is still inferior to our original model, which demonstrates the irreplaceability of the dynamic attention mechanism.
>
> - **Conclusion**: In the dynamic attention module, GRU can generally be replaced by LSTM if efficiency is not a concern. However, at the overall module level, the dynamic attention mechanism cannot be effectively substituted by other types of attention.
> These findings have been added to the revised manuscript.
>
> [6] Hochreiter S and Schmidhuber J. "Long short-term memory". Neural computation 1997.
> [7] Vaswani A, et al. "Attention is all you need". NIPS 2017.
> [8] Gu A, Dao T. "Mamba: Linear-time sequence modeling with selective state spaces". arXiv 2023.
>
> ---
>
> ###  **Weakness 1**: Redundant definition.
>
> Thank you for your valuable suggestion. We have revised the definition of heterogeneous temporal graph for clarity. Specifically, it is now defined as: $\mathcal{G}=(\\{G^{t}\\}_{t=1}^{T} )$, where $G^{t}$ denotes a heterogeneous graph at time $t$.
>
> ---
>
> ###  **Weakness 2 3**: More clarification on contributions.
>
> Thank you for your comments. We would like to clarify that the primary motivation of our work is to promote a simple yet efficient method for heterogeneous temporal graph. Therefore, the design of our LLM-enhanced module adheres to this principle: while more sophisticated LLM-based methods may improve performance, they introduce significant computational overhead.
> Our method strikes a balance between efficiency and effectiveness, achieving notable performance gains with minimal overhead.
> We compared the training time overhead introduced by LLMs in existing LLM-based methods. The results are shown below.
>
> | LLM time cost | OGBN_MAG | AMiner | YELP | COVID19 |
> |---|---|---|---|---|
> | CasMLN [5] | 1.33× | 1.31× | 1.74× | 1.37× |
> | SE-HTGNN (ours) | 1.06× | 1.04× | 1.03× | 1.04× |
> |  |  |  |  |  |
>
> The results indicate that CasMLN’s LLM-based techniques incurs a training time overhead of 1.31 to 1.74× compared to not using LLM. Meanwhile, our LLM-enhanced prompt module only introduces a marginal increase of 1.03 to 1.06×. These findings highlight the superior cost-effectiveness of our approach.

---

> > ### Comment · Reviewer_FjxQ · 2025-08-01
> >
> > Thank you for the detailed response. After reviewing your answers and the results provided, I would like to raise my rating to 4.

---

> ### Author Response · Authors · 2025-08-01
>
> Thank you for your positive feedback. We sincerely appreciate your review, which helps us improve our paper.

---

### Official Review · Reviewer_heSU · 2025-07-03

**Clarity:** 3
**Significance:** 3
**Originality:** 3
**Rating:** 5
**Confidence:** 4

**Summary:**

This paper proposes a novel framework called SE-HTGNN (Simple and Efficient Heterogeneous Temporal Graph Neural Network) for learning on heterogeneous temporal graphs. The authors identify limitations in existing approaches, such as the decoupled modeling of spatial and temporal information and increasing model complexity. SE-HTGNN addresses these issues by integrating temporal modeling into spatial learning through a dynamic attention mechanism that leverages historical attention coefficients. In addition, the model incorporates large language models (LLMs) to enhance node type representation with external knowledge. Experiments on multiple real-world datasets demonstrate that SE-HTGNN achieves state-of-the-art performance with improved efficiency compared to previous methods.

**Questions:**

**Q1:** Could the authors provide more detailed empirical analysis or discussion regarding the scalability and memory efficiency of SE-HTGNN, particularly as the number of node/relation types or the temporal window length increases? How does the model behave in terms of memory and runtime on truly large-scale heterogeneous temporal graphs?

**Q2:** The ablation study on the dynamic attention mechanism mainly focuses on removing it versus keeping it. Have the authors considered comparing with alternative temporal fusion strategies or conducting more fine-grained analysis (e.g., on the evolution of attention coefficients for different relation types or across varying time scales)?


**Q3:** Can the authors clarify the practical trade-offs introduced by LLM-based node type initialization? Specifically, what is the additional computational and memory overhead, and how sensitive is the model’s performance to the choice or size of the LLM? How would the system handle cases where node types change or new types are introduced over time?

**Ethical Concerns:**

["NO or VERY MINOR ethics concerns only"]

**Final Justification:**

The authors have provided substantial additional experiments and clarifications addressing my earlier concerns. They demonstrated clear scalability and memory efficiency advantages on large-scale and long-sequence datasets, performed broader ablations on the dynamic attention mechanism, and analyzed its evolution behavior in detail. The trade-offs of LLM-based initialization were quantified, showing minimal overhead and flexibility across models. These additions strengthen both the empirical and practical validity of SE-HTGNN. Given the thorough rebuttal and convincing new evidence, I raise my rating from borderline accept to accept.

**Limitations:**

See Weakness 1 - Weakness 4.

**Paper Formatting Concerns:**

There is no problem associated with paper formatting.

**Quality:**

3

**Strengths And Weaknesses:**

**Strengths:**

1. **Unified Spatio-Temporal Modeling:** The paper introduces a unified framework (SE-HTGNN) that combines spatial and temporal modeling through a dynamic attention mechanism, addressing the common issue of decoupled learning stages in prior work. This design is conceptually simple and helps to reduce model complexity.

2. **Dynamic Attention Mechanism:** By retaining and propagating historical attention coefficients across time steps, the proposed dynamic attention mechanism enables more effective modeling of long-term dependencies in heterogeneous temporal graphs. This approach helps to improve representation quality and model stability.

3. **Incorporation of LLM-based Prior Knowledge:** The model leverages large language models (LLMs) to encode node type semantics as additional prior knowledge, which provides a novel angle for injecting external information into graph neural network training and potentially improves adaptability to complex node types.

4. **Strong Empirical Results and Efficiency:** Extensive experiments on several benchmark datasets show that SE-HTGNN consistently outperforms previous state-of-the-art methods on both accuracy and efficiency metrics. The reported speed-up (up to 10× on some tasks) and better convergence rates make the proposed method attractive for practical applications on large-scale heterogeneous temporal graphs.

**Weaknesses:**

1. **Lack of Analysis on Scalability and Memory Efficiency in Practice.** Although the paper claims improved efficiency and reduced complexity, there is limited discussion or empirical analysis on the scalability of SE-HTGNN for truly large-scale heterogeneous temporal graphs in real industrial settings. For example, memory usage with increasing node/relation types or longer temporal windows is not systematically studied.

2. **Ablation on Dynamic Attention Mechanism is Limited.** The dynamic attention mechanism is central to the claimed improvement, but the ablation study mainly compares removing it versus keeping it. There is little exploration of alternative temporal fusion methods or variants, nor is there a fine-grained analysis of the attention evolution behavior on different relation types or time scales.

3. **LLM-based Initialization Introduces Unclear Tradeoffs.** The integration of LLM embeddings for node type prompts is interesting, but the paper does not analyze the trade-off between the potential benefit and the additional preprocessing cost or memory overhead. In practice, using large LLMs for embedding extraction may be costly, especially if the node type set changes or expands dynamically.

4. **Limited Insight into Model Interpretability and Decision Process.**
While the model introduces mechanisms for incorporating historical attention, there is minimal discussion or visualization on what the model is actually learning about spatio-temporal dependencies or heterogeneous relations. More qualitative analysis or case studies (e.g., attention heatmaps, error analysis) would improve understanding of the model's behavior and potential failure modes.

---

> ### Author Rebuttal · Authors · 2025-07-31
>
> Dear Reviewer heSU, we are deeply grateful for your careful reading and valuable feedback of our work. We hope to have carefully addressed your concerns and made updates to the manuscript based on your feedback. Here is our response to your comments and questions.
>
> ---
>
> ###  **Question 1**:  Discussion regarding the scalability and memory efficiency of SE-HTGNN.
> Thank you for you suggestions, we conducted additional experiments to further address your concerns.
>
> - **Memory**:  we compare the GPU memory usage (in GPU MB) of different methods as the temporal window increases, using two challenging settings: a **large-scale graph** with millions of edges (OGBN-MAG) and a long-sequence dataset (Covid19).
> The detailed results are summarized below.
> |OGBN_MAG (Temporal window): |2|3|4|5|6|7|
> |:---:|:---:|:---:|:---:|:---:|:---:|:---:|
> |HTGNN|2622|5249|8719|13084|18131|OOM|
> |DyHART|3648|7266|12067|18091|OOM|OOM|
> |CasMLN|5058|6540|8014|9587|11038|12597|
> |Ours|**1134**|**2110**|**3440**|**5094**|**6585**|**8036**|
> ||||||||
>
> | Covid19 (Temporal window): |3|7|14|21|28|60|
> |:---:|:---:|:---:|:---:|:---:|:---:|:---:|
> | HTGNN | 24 | 58 | 119 | 185 | 255 | 602 |
> | DyHART | 40 | 82 | 159 | 238 | 361 | 856 |
> | CasMLN | 86 | 157 | 196 | 210 | 235 | 354 |
> | Ours |**24**|**48**|**86**|**114**|**149**|**238**|
> ||||||||
>
> Conclusion: the experimental results show that our method consistently maintains the lowest memory usage, regardless of whether the dataset is large-scale or contains long temporal sequences. Specifically, our approach exhibits approximately linear growth in memory consumption as the time window increases, which is mainly attributed to the length of sequence, without introducing extra model parameters. In contrast, DyHART and HTGNN encounter out-of-memory (OOM) issues on large graphs, which demonstrating the high model complexity.
>
> - **Runtime**: we compare the training time (in GPU second) of different methods as the temporal window increases, using a long-sequence dataset (Covid19).
> | Covid19 (Temporal  window): |7|14|30|
> |:---:|:---:|:---:|:---:|
> | CasMLN  | 637 | 752 | - |
> | Ours |**64**|**86**|**143**|
> |||||
>
> Conclusion：Since CasMLN fails to converge when the temporal window is set to 30, we stop the comparisons with larger window sizes. For our method, the training time does increase moderately with a larger temporal window, but it remains significantly lower than that of the most efficient baselines and it further reduce the MAE metric from 497 to 462.
>
>
> - **Number of node/relation types.**  While type changing setting is uncommon in HTG benchmarks, we provide a brief comparison to illustrate scalability. Specifically, when our method encounters new relation or node types, the additional parameter overhead comes only from introducing a new GRU for computing attention coefficients. This is more scalable than previous methods, which often require adding a set of graph attention network (GAT) parameters for each new type.
>
> ---
>
> ###  **Question 2**:  Provide more ablation study and analysis on Dynamic Attention Mechanism.
> Thank you for your constructive suggestions, we add two ablation experiments: (1) GRU variants and  (2) entire attention mechanism variants.
>
> - **For the replacement of GRU**,  we replace the GRU with other sequence-based modules as follows: **LSTM** [1], **Transformer** [2], and **Mamba** [3]. It is worth noting that the Transformer does not maintain a hidden state over time, and therefore cannot benefit from the LLM-enhanced prompt module. The results are summarized below.
> | Alternative  | OGBN_MAG |  | AMiner |  | YELP |  | COVID19 |  |
> |---|---|---|---|---|---|---|---|---|
> |  | AUC | AP | AUC | AP | MacroF1 | Recall | MSE | RMSE |
> | LSTM | 92.77±0.48 | 92.28±0.45 | 90.52±0.48|89.87±0.58|**44.31±0.97** | **44.73±0.52** | 506±7 | 1078±16 |
> | Mamba | 91.79±1.32 | 90.92±1.45 | 87.81±1.65 | 87.12±1.59 | 41.45±2.41 | 41.58±3.10 | 601±67 | 1259±84 |
> | Transformer | 90.75±0.32 | 89.93±0.34 | 90.93±0.45 | 89.89±0.32 | 43.59±1.21 | 43.64±1.19 | 529±12 | 1098±23 |
> | Original (GRU) | **93.13±0.56**  | **92.71±0.52**  | **91.08±0.59**  | **90.03±0.48**| 44.24±0.88  | 44.68±0.43 | **497±5** | **1069±11** |
> ||||||||||
>
> Conclusion: according to the experimental results, the original model (GRU-based) and the LSTM-based variant achieve comparable performance. However, GRU, as a simplified version of LSTM, offers better computational efficiency.  In contrast, both Transformer-based and Mamba-based variant underperform in this task. For Mamba model, as reported in its original paper,  it suffers from unstable training, which may account for its inferior performance.
>
> - **For the replacement of entire attention,** we replaced the original dynamic attention-based fusion module with self-attention and gate-attention. The experimental results are summarized as follows.
> | Ablation | OGBN_MAG |  | AMiner |  | YELP |  | COVID19 |  |
> |---|---|---|---|---|---|---|---|---|
> |  | AUC | AP | AUC | AP | MacroF1 | Recall | MSE | RMSE |
> | Self-attention | 91.65±1.22 | 90.65±1.14 | 88.73±0.82 | 88.21±0.94 | 42.41±1.32 | 42.73±0.97| 545±33|1114±42|
> | Gate-attention | 87.94±2.42 | 87.24±1.89 | 87.42±1.24 | 86.55±1.33 | 38.96±3.27 | 39.26±2.27| 574±45| 1216±68|
> | Original |**93.13±0.56** |**92.71±0.52** | **91.08±0.59** | **90.03±0.48** | **44.24±0.88** | **44.68±0.43**| **497±5**| **1069±11** |
> |  |  |  |  |  |  |  |  |  |
>
> Conclusion: from the results, the performance of variants is still inferior to our original model, which demonstrates the irreplaceability of the dynamic attention mechanism.
>
> - **Analysis of the attention evolution behavior**: we provided a visualization of the attention coefficients in Figure 5 of the manuscript. Here, we offer a more detailed analysis.
> Figure 5 (2) illustrates how attention coefficients evolving over a 30-day period for a specific U.S. state in the COVID-19 dataset. In this figure:
>     - The blue "state" denotes a specific state node.
>     - The orange "state" refers to **all** neighboring state-type nodes of the blue state.
>     - The "county" refers to **all** county-type nodes governed by the blue state.
>     - The numbers reflect the trend of attention coefficients changes over time.
>
> In the task of predicting the new cases of COVID-19 in that state, we observe that the attention coefficients assigned to neighboring states are consistently higher than those for county. This suggests that the state's case increases are likely more influenced by adjacent states.
> Furthermore, the coefficients of the state-type node increases from 0.58 to 0.61, indicating that recent information from the neighboring state becomes more important than historical data.
> This observation also supports our insight that historical attention patterns are useful for future learning. Specifically, in this case, we see that state-type deemed important in historical snapshot retain or even increase their importance later on.
>
> We have included this interpretation in the revised manuscript and provide more qualitative analysis and case studies to improve clarity and readability.
>
> [1] Hochreiter S and Schmidhuber J. "Long short-term memory". Neural computation 1997.
> [2] Vaswani A, et al. "Attention is all you need". NIPS 2017.
> [3] Gu A, Dao T. "Mamba: Linear-time sequence modeling with selective state spaces". arXiv 2023.
>
> ---
>
> ###  **Question 3**:  clarify the practical trade-offs introduced by LLM-based node type initialization?
> Thank you for your constructive suggestions.
>  - **Additional computational and memory cost**: During the **preprocessing** stage, we take the Covid19 dataset and use the LLaMA3-8B model to generate prompts as an example. On an RTX 3090, this process takes only **3.98** seconds and consumes approximately 17 GB of GPU memory—almost identical to the overhead reported in CasMLN [4].
> During the **training** stage, since the GPU memory usage remains almost unchanged, we compared the training time overhead introduced by LLMs. The results are shown below.
> | LLM time cost | OGBN_MAG | AMiner | YELP | COVID19 |
> |---|---|---|---|---|
> | CasMLN [5] | 1.33× | 1.31× | 1.74× | 1.37× |
> | SE-HTGNN (ours) | 1.06× | 1.04× | 1.03× | 1.04× |
> |  |  |  |  |  |
>
> The results indicate that CasMLN’s LLM-based techniques incurs a training time overhead of 1.31 to 1.74× compared to not using LLM. Meanwhile, our LLM-enhanced prompt module only introduces a marginal increase of 1.03 to 1.06×. These findings highlight the superior cost-effectiveness of our approach.
>
> - **Sensitive to different LLM**: We compare several different LLMs, as shown below.
> |  | OGBN_MAG |  | AMiner |  | YELP |  | COVID19 |  |
> |---|---|---|---|---|---|---|---|---|
> | |AUC|AP|AUC|AP|MacroF1|Recall|MSE| RMSE|
> | llama2−7B | 92.25±1.08 | 92.25±1.08 | 89.92±0.64|89.76±0.62|42.84±1.08|43.20±0.82|512±12|1098±26|
> | GPT−3.5 | 92.85±0.98 | 92.54±0.98  | 90.53±0.51 | 90.26±0.42 | 43.37±1.23 | 43.97±0.88 | 504±8 |1081±19|
> | GPT-4 | **93.38±0.46** | **93.12±0.45** | 91.24±0.47 | 90.15±0.41 | 43.98±0.65 | 43.45±0.58 | 493±5 |1056±13|
> | DeepseekV3 | 93.31±0.42 | 92.98±0.45 | **91.32±0.32** | **90.48±0.38**| **44.48±0.65** | 44.67±0.46|**492±7**|**1052±18**|
> | llama3−8B | 93.13±0.56|92.71±0.52|91.08±0.59|90.03±0.48|44.24±0.88|**44.68±0.43**|497±5|1069±11|
> ||||||||||
>
> Experimental results show that larger models can generate more informative initialization, leading to better performance. To balance efficiency and effectiveness, we adopt LLaMA3-8B in practice.
>
> - **How to handle new type**: In most HTG applications, changes typically occur in structure or features. If new types emerge, all existing models generally require retraining or continual learning. Therefore, the LLM processing time for new types becomes negligible compared to the overall training time. We appreciate your suggestion and will consider this aspect in our future work.
>
> [4] Wang F, et al. "LLM-enhanced Cascaded Multi-level Learning on Temporal Heterogeneous Graphs" ACM SIGIR 2024.
>
> ---

---

### Note · Authors · 2025-08-14

Dear SAC, AC, and Reviewers,
We would like to express our gratitude for your contributions to the community. We are grateful that all reviewers provided positive scores for our paper. Your insightful comments have helped us further improve the quality of our paper. Below, we briefly summarize our paper and the discussion process.

This manuscript focuses on two issues in heterogeneous temporal graph (HTG) learning: **(1)** The low efficiency, and **(2)** How to leverage LLMs to improve HTG learning. To this end, we propose a simple yet efficient new learning paradigm for HTGs. Experiments show superior of our method, with up to **10×** speedup over SOTA.

During the discussion, we first express our gratitude for reviewers' recognition of the paper’s **strengths**:

- Novel learning paradigm unifying temporal and spatial modeling, improving efficiency (All reviewers). Reviewer YMqu noted that this is “a good innovation for practical usage on large graphs.”
- Innovative integration of LLMs and GNN to boost domain understanding (Reviewer heSU, FjxQ, KYMR). Reviewer heSU highlighted this as “a novel angle for injecting external information into graph.”
- The experiment results are sound, giving evidences from various aspect to justify the improvement of proposed method (Reviewer YMqu).

In addition, the reviewers raised several **concerns**, which we summarize below together with our **responses**:

- More ablation studies. In response, we conducted additional ablation study as suggested. Specifically, we replaced the GRU with other sequence models (Reviewer heSU, KYMR), replaced the entire dynamic attention with other attention (Reviewer heSU, FjxQ). In addition, we included ablation study on the initialization of the GRU (Reviewer KYMR) and on the simplified aggregation (Reviewer FjxQ).

- Analysis on scalability and memory efficiency (Reviewer heSU, KYMR). In response, we compared the changes in memory usage and training time as the temporal window increased, showing that we maintain the lowest overhead. We also analyze the trade-off between benefit and the additional cost of LLMs.

- More benchmark (Reviewer KYMR, YMqu). We added experiments on an additional benchmark.

- More discussion on model's behavior (Reviewer heSU, YMqu). In response, we provided a detailed explanation and analysis of the dynamic attention through a case study.

We once again thank the reviewers and chairs for their valuable time.

Best regards,

Authors of paper 28073

---

### Decision · Program_Chairs · 2025-09-17

**Decision:**

Accept (poster)

**Comment:**

This paper focuses on HTG learning in terms of efficiency and LLM incorporation. Reviewers confirm the merits of the paper including effective spatial and temporal integration, nice LLM and GNN combination and strong empirical results. While initial reviews raised questions regarding ablation studies, novelty insufficiency, analysis comprehensiveness, the author rebuttal has adequately addressed the concerns. I hence recommend accepting the paper.